# Mechanical plasticity of collagen directs branch elongation in human mammary gland organoids

B. Buchmann[1,4], L. K. Engelbrecht[1,2,4], P. Fernandez[1], F. P. Hutterer [1], M. K. Raich[1], C. H. Scheel [2,3✉] & A. R. Bausch [1✉]

Epithelial branch elongation is a central developmental process during branching morphogenesis in diverse organs. This fundamental growth process into large arborized epithelial networks is accompanied by structural reorganization of the surrounding extracellular matrix (ECM), well beyond its mechanical linear response regime. Here, we report that epithelial ductal elongation within human mammary organoid branches relies on the non-linear and plastic mechanical response of the surrounding collagen. Specifically, we demonstrate that collective back-and-forth motion of cells within the branches generates tension that is strong enough to induce a plastic reorganization of the surrounding collagen network which results in the formation of mechanically stable collagen cages. Such matrix encasing in turn directs further tension generation, branch outgrowth and plastic deformation of the matrix. The identified mechanical tension equilibrium sets a framework to understand how mechanical cues can direct ductal branch elongation.

[1] Lehrstuhl für Biophysik E27, Physics Department and Center for Protein Assemblies CPA, Technical University Munich (TUM), Garching, Germany. [2] Institute of Stem Cell Research, Helmholtz Center for Health and Environmental Research Munich, Neuherberg, Germany. [3] Department of Dermatology, St. Josef Hospital, Ruhr-University Bochum, Bochum, Germany. [4]These authors contributed equally: B. Buchmann, L. K. Engelbrecht. ✉email: scheelch@me.com; abausch@mytum.de

Branching morphogenesis leads to the formation of a tree-like, ductal epithelium in diverse organs such as kidney[1], lung[2], pancreas[3] or mammary gland[4] and is known to be the result of an interplay between chemical signals and mechanical cues[5,6]. Especially, matrix remodeling processes affecting stiffness, density and composition of the ECM play a central part during development and tumorigenesis[7–9]. In particular, structural and mechanical properties of the surrounding ECM seem to act as guiding cues for branch elongation[10–12]. However, many aspects of the mechanisms guiding collective outgrowth of epithelial structures have yet to be identified[13–15], which is further complicated by the fact that branching morphogenesis differs greatly between tissues due to organ-specific mechanical and chemical environments[5,16]. Two-dimensional model systems have been instrumental to identify possible underlying self-organization processes[17]. Specifically, it has been shown that outgrowth of epithelial cell layers during expansion is orchestrated by a balance of tight coupling between cells and their adhesion to the substrate, ultimately leading to the generation of tension within the layer[15]. In addition, strong cell-cell contacts enable large-scale interactions that result in directed collective cell migration depending on confinement conditions[18]. Experiments in a three-dimensional environment have highlighted the role of aligned collagen fibers and dynamic cell-matrix interaction as guiding cues for epithelial outgrowth[11,12,19]. Such structural changes can be induced by mechanical forces exerted by individual cells embedded in collagen gels and in turn can affect their migration behavior[20,21]. Yet, the origin of large-scale collagen deformations as observed during collective epithelial expansion remains a topic of intense research. Further, the lack of in vivo relevant assays hampers the analysis of self-organizational principles driving branching morphogenesis in the human mammary gland.

In this work, we used mammary organoids derived by embedding single primary human basal cells into floating 3D-collagen type I gels. Thereby, we were able to observe plastic collagen fiber alignment, collagen accumulation, tension generation and, ultimately branch elongation through invasive outgrowth as one integrated, dynamic process. In this single-cell-based assay, it has been shown that endogenous contractility of the basal cells, indicated by whole-gel shrinkage of the freely floating collagen gels, is required for the growth of Terminal Ductal Lobular Unit (TDLU)-like organoids[22]. Thus, by tracking internal cellular dynamics we investigated the functional role of tension generation during branch elongation and correlated the resulting matrix remodeling with the observed collective cell migration. Thereby, we linked the observed local deformations to the mechanical plasticity of the collagen I matrix, which ultimately resulted in the formation of a collagen cage encasing the growing organoid branches. Such an encasing is similar to collagen densification observed around mammary ducts in vivo[11,23], which is thought to arise from collagen secretion and remodeling mediated by macrophages and fibroblasts within the stroma[23]. However, based on our model, this collagen densification can further be facilitated by epithelial self-organization.

## Results

**Human mammary gland organoids invade the ECM by non-continuous contractions.** For the generation of human mammary organoids, single EpCAM$^+$/CD49f$^+$/CD10$^+$ basal mammary epithelial cells were isolated from human reduction mammoplasties and seeded at clonal density into freely floating collagen type I gels[22] (Fig. 1a, Supplementary Fig. S1A). The development of the mammary organoids can be classified in three stages (Fig. 1b): the establishment phase (day 1–6), the branch

elongation phase (day 7–9) and the alveologenesis phase (day 10–14). Characteristic for the establishment phase were small, rod-like cell clusters with lengths of around 120 μm (Fig. 1c), that showed only rudimentary branches. During the elongation phase, these branches invaded into the ECM and developed primary and secondary side branches. During branch elongation, we detected proliferative cells throughout the whole organoid (Supplementary Fig. S1B) and observed the formation of filopodia-like cellular protrusions at the leading edge of the branches, hinting towards an invasive branching process (Supplementary Fig. S1C). Ultimately, during the alveologenesis phase, the organoids reached a size of around 1 mm in diameter (Fig. 1c). Moreover, the branches of these organoids thickened and formed rounded end buds allowing the formation of a lumen, resulting in the formation of TDLU-like structures. This third phase also coincided with polarized expression of the previously described markers for the two major lineages within the mammary gland[22]: p63 for basal cells, expressed within the outer cell layer adjacent to the ECM, and GATA3 for luminal cells, expressed within the cell layer adjacent to the forming lumen (Supplementary Fig. S1D). The expression of these specific markers showed that the mammary organoids generated in 3D floating collagen I gels resemble the bilayered architecture of the human mammary gland.

The expansion of the developing organoids was observed by live-confocal-microscopy over extended time periods, from hours to days (Supplementary Video 1). As observed by means of embedded tracer particles, the elongation of each branch induced a significant long-range deformation field within the ECM directed towards the branch (Fig. 1d). After 14 days of culture these local deformations added up to millimeters, which led to a macroscopic shrinkage of the collagen gel to about half its diameter[22]. In the near-field, the observed strains exhibited local heterogeneities and were highly anisotropic: deformations were the strongest directly at the tip of the branch in extension to the direction of elongation and steadily decreased with increasing angle from the branch tip (Fig. 1e/ Supplementary Fig. S1E). Moreover, we observed a periodic displacement of the embedded tracer particles towards the branch with periods in the hour scale (Fig. 1f). Specifically, bead displacements towards the branch were followed by relaxations in the opposite direction. These periodic displacements were not restricted to the close proximity of the branches but could also be detected further away from the organoid (Fig. 1g). Computing the cumulative displacement of the beads over time showed a steady increase in bead displacement towards the branches (Fig. 1h, red line).

We concluded that the endogenous contractility of the myoepithelial cell sheet specifically induces large anisotropic deformation fields in the ECM in front of the elongating branch. Indeed, already during the organoid establishment phase, we observed that small clusters consisting of just a few basal cells were already able to induce considerable deformation of the surrounding ECM (Supplementary Fig. S1F, H), whereas non-contractile luminal mammary epithelial cells grew as multicellular spheres driven by proliferative pressure without resolvable ECM deformation (Supplementary Fig. S1G, H). Occasionally, only short-ranged and localized elastic deformations appeared. To test directly whether endogenous contractility of basal cells is required for branching initiation, the Rho kinase (ROCK)-inhibitor Y-27632 was added to the culture medium from day 1 on. Indeed, branch formation was inhibited, leading mainly to the emergence of unstructured and dense cell clusters as observed previously[22] (Supplementary Fig. S1I). Moreover, the inhibition of cellular contractility during the branch elongation phase on day 10 prevented further branch elongation as well as the global anisotropic contraction of the ECM as observed before. Whereas the formation of filopodia-like-protrusions was only observed in

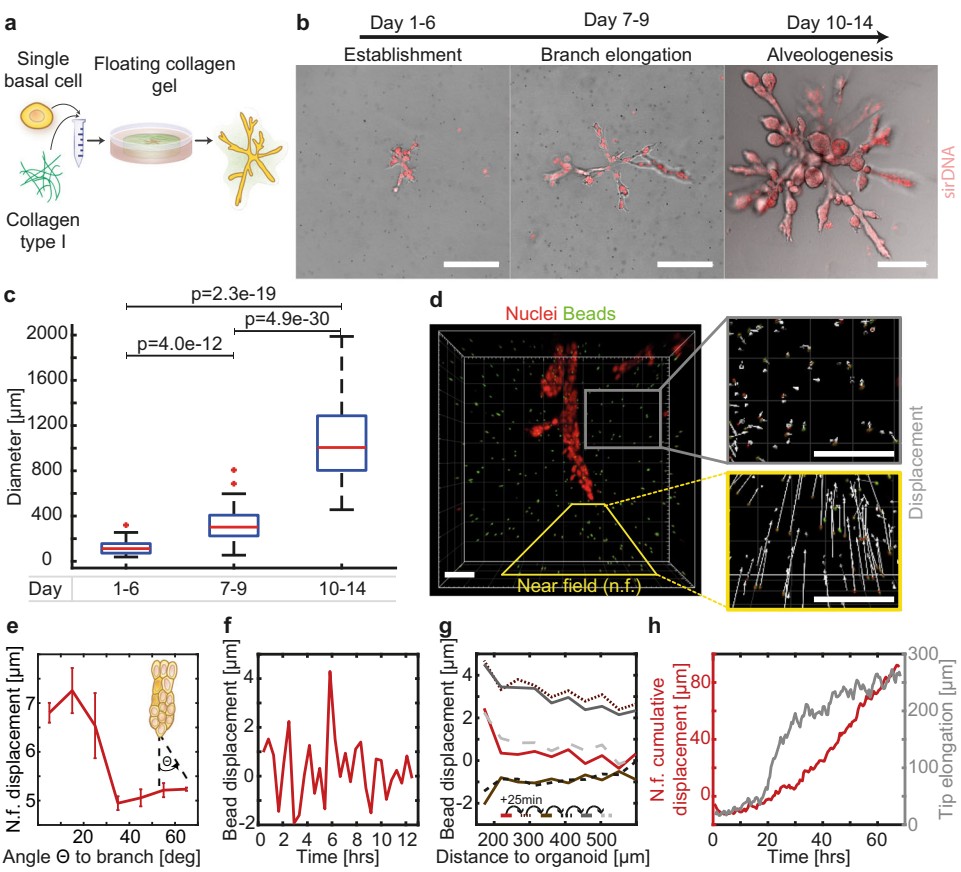

**Fig. 1 Human mammary gland organoids invade the ECM by non-continuous contractions. a** Schematic overview of 3D culture: single primary human basal mammary epithelial cells are cultured in floating collagen gels. **b** Characteristic organoid morphology at three developmental stages (Establishment $n = 36$ organoids, Branch elongation $n = 75$ organoids, Alveologenesis $n = 111$ organoids). Nuclei are visualized using sirDNA. **c** The organoid diameter of the long axis during the different stages reveals an increase in diameter during the elongation phase (Establishment $n = 36$ organoids, Branch elongation $n = 75$ organoids, Alveologenesis $n = 111$ organoids). Box plots indicate median (red line), 25th, 75th percentile (blue box) and 5th and 95th percentile (whiskers) as well as outliers (single points). **d** Live-cell imaging reveals an anisotropic deformation field with strong deformations in front of the branches and no deformation at the sides of the branches ($n = 24$ organoids). The near field (n.f.) is defined as area between the branch tip and the ECM 300 μm away from it. **e** The deformation is decreasing with increasing angle θ to the branch ($n = 14$ organoids). Error bars, mean ± s.d. **f** The bead displacement is non-continuous over time with contractions towards the branches and relaxations into the opposite direction ($n = 14$ organoids). **g** ECM contractions and relaxations slowly diminish with increasing distance to the organoid. Between each line 25 min passed, highlighting the alternations between contractions of the ECM towards the branches and relaxations away from them ($n = 23$ organoids). **h** The cumulative bead displacement in front of branches is increasing over time (red, $n = 5$ organoids), while the branch elongation is discontinuous in time (gray, $n = 7$ organoids). Scale bars, 200 μm (**b**), 70 μm (**d**). Organoids were derived from three biologically independent donors (Supplementary Table S1). P values are from a two-tailed Mann–Whitney test and provided in Supplementary Table S5. Source data are provided as a Source Data file.

leading cells in control organoids (Supplementary Fig. S1J), ROCK-inhibitor treated organoids displayed cellular protrusions in stalk cells all along the branch axis (Supplementary Fig. S1K, Supplementary Video 2). This observation correlated with deformations of just a few microns perpendicular to the branch in contrast to the large oriented deformations observed in control organoids that emanated from the leading front in direction of branch elongation.

Taken together, we observed that during branch elongation a highly anisotropic deformation field was generated as a result of endogenous contractility. Importantly, ECM deformations were found to be non-continuous and long-ranging. In addition, branch elongation occurred discontinuously in time, showing a back-and-forth movement by the leading cells (Fig. 1h, gray line). Together, these observations hinted towards dynamic cellular rearrangements within elongating branches.

**Collective cell migration facilitates ECM deformations.** In order to identify the mechanism driving the discontinuous

branch elongation, we studied cellular dynamics within elongating branches by live-cell imaging of organoids expressing LifeAct-GFP. Thereby, we could observe that the tip of the branch was led by a few cells, which formed filopodia-like protrusions and appeared to actively invade into the collagen network (Fig. 2a, Supplementary Video 3). Those leader cells were followed by stalk cells, which were reported previously to support branch elongation via intercalation processes as observed in epithelial multilayers[24], during endothelial sprouting[25] or tube elongation of the Drosophila trachea[26]. However, nuclei labeling using sirDNA revealed a dynamic exchange of leader cells within elongating organoid branches. More specifically, leading cells were regularly overtaken by cells behind them which thereafter established the new tip of the branch, a phenomenon previously described during endothelial sprouting[27,28] (Fig. 2b, Supplementary Video 4). More than 70% of the observed tip exchanges were initiated by a deceleration of the tip cells, while follower cells were continuously migrating outward (Supplementary Fig. S2A, B). The former leading cells became part of the stalk cells and

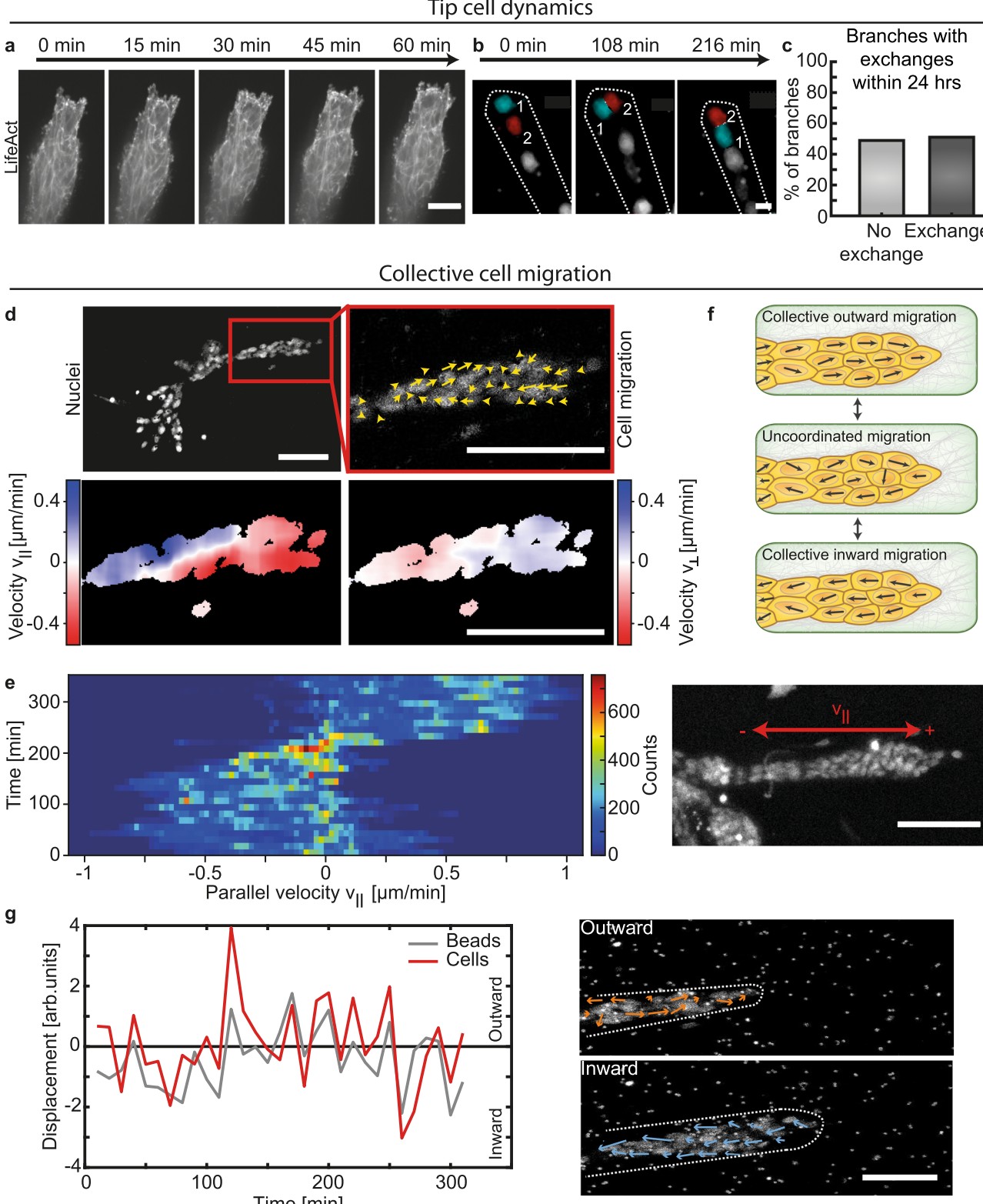

either stayed behind the new leading cell or, in rare cases, integrated into the cellular motion within the branch ($n = 3$). In a time course of 24 h, in 51% of analyzed branches ($n = 47$) at least one exchange of a leader cell was observed (Fig. 2c). Moreover, we observed those tip cell exchanges to happen more likely in shorter and narrower branches (Supplementary Fig. S1C, D).

We next analyzed cellular dynamics in the whole organoid by nuclear labeling. Cells throughout the whole organoid were highly motile and migrated in a collective manner, leading to cells migrating in cohorts (Supplementary Video 1). We could observe that those collectively migrating cohorts changed their size over time from only a few cells migrating in the same direction to

**Fig. 2 Collective cell migration facilitates ECM deformations. a** LifeAct-GFP staining reveals dynamic remodeling of the actin network in the leading cells. Invadopodia of leader cells show dynamic interaction with the ECM throughout the invasion process. **b** During branch elongation leading cells (red cell, 2) exchange places with cell behind them (cyan cell, 1). **c** Within 24 h tip cell exchange was observed in about half of the branches analyzed ($n = 47$ organoids). **d** Internal collective cell migration ($n = 16$ organoids). Top panel: Nuclei channel representing the organoid morphology (left). Total velocity measurement of the cells inside the organoid reveals highly dynamic cells (right). Low panel: Velocity in parallel $v_{||}$ (left) and orthogonal $v_{\perp}$ direction (right) shows clusters of cells collectively moving in the same direction. **e** Cell velocity distribution within one branch of an organoid at day 12 over time. Only the velocity parallel to the branch is plotted. Signs are defined as depicted in the according nuclei channel. **f** Schematic overview of observed collective cell migration phases. **g** Bead and cell motion are both discontinuous in time and show periods of correlated phases, during which their direction is pointing in the same direction ($n = 11$ organoids). During highly correlated phases, outward pointing cell migration correlates with relaxations of the beads in front of the branch away from it. During inward pointing cell migration beads get pulled towards the organoid. Confocal shows a representative organoid at day 8. Scale bars, 100 μm (**a, d, e, g**), 10 μm (**b**). Organoids were derived from three biologically independent donors (Supplementary Table S1). Source data are provided as a Source Data file.

persistent movements within whole branches. Specifically, cells exhibited a movement predominantly parallel to the branch axis with only smaller movements in the orthogonal direction (Fig. 2d). Particularly within single branches, stalk cells migrated with speeds of up to 1 μm/min, but frequently changed their direction or paused their movement (Fig. 2e). Cells within single branches exhibited different phases of cell migration. Phases of cells moving collectively in the same direction within whole branches were followed by phases in which cells moved individually, resulting in a temporary loss of coordination. Ultimately, the uncoordinated movement changed to a collective migration phase again (Fig. 2f).

Importantly, the direction of the net movement of the cells was synchronous with the deformation field within the ECM in front of the branch tips (Fig. 2g). Thus, we could observe phases in which cell movement and ECM deformation were highly correlated. When collective cell motion was pointing outward, mainly relaxations in the ECM were observed. In contrast, when stalk cells within the whole branch moved away from the tip towards the organoid center, large deformations within the ECM towards the tip appeared. No directed collective cell movements and ECM deformations were observed perpendicular to the long axis of the branches (Supplementary Figs. S1E and S2E). Taken together, these observations strongly suggested that the non-continuous ECM deformations in front of elongating branches result from the collective nature of cell movements within the branch and did not emanate from the tip cells only. Indeed, such long-range deformation fields could not be recapitulated in experiments with single cells (Supplementary Fig. S3A).

**ECM deformations are enabled through tension equilibrium.** To further investigate the mechanism of the underlying force buildup within the branch, we performed immunofluorescence staining of alpha smooth muscle actin. Indeed, we observed high expression of alpha smooth muscle actin in basal cells at the outer cell layer of branches adjacent to the ECM (Supplementary Fig. S3B). Moreover, phalloidin staining revealed thick actin cables connecting neighboring cells (Supplementary Fig. S1C). In addition, we detected strong cell coupling of the cells within the branches via E-cadherin, supporting that tension buildup observed during live cell imaging resulted from a cell-collective effort (Supplementary Fig. S3C). To investigate this further, the actin network was disrupted by addition of Cytochalasin D. This led to a loss of tension, resulting in an instantaneous relaxation of the organoid branches. Due to the elastic counterforce of the surrounding ECM, relaxation was reflected in concomitant branch expansion (Fig. 3a, b, Supplementary Video 5). Yet, the ECM structure in front of the invading branches was unaffected by the Cytochalasin D treatment (Fig. 3c). In order to further investigate the origin of the tensile forces inside the ECM, UV laser ablation of the collagen matrix in front of invading branches

was conducted (Fig. 3d). Such ablation was followed by an instantaneous relaxation of the whole branch towards the organoid body (Fig. 3e, Supplementary Video 6). Accordingly, the ECM relaxed in the opposite direction of the initial deformation field, suggesting that the tension in the ECM originated from tension buildup within the organoid branches.

To prove the necessity of collective tension buildup, we lowered cell-cell adhesion by addition of a function-blocking anti-E-cadherin antibody (HECD1 clone) throughout the entire period of organoid culture. As a result, thin stick-like branches evolved, that completely lacked alveologenesis. (Fig. 3g, Supplementary Fig. S3E–G). Further, the initial collective cell migration observed in control organoids was completely abolished, resulting in only individual short-ranging cellular movements (Supplementary Fig. S3H, Supplementary Video 7). Complementary, inhibiting endogenous contractility of the basal cells by the addition of the ROCK-inhibitor Y-27632 throughout the entire period of organoid culture led to the formation of disorganized star-like multicellular structures, which did not resemble the branched architecture of control organoids (Fig. 3g, Supplementary Fig. S1I). Taken together, these results suggested that tension is generated by a collective of contractile cells within the branch and is equilibrated by the surrounding ECM (Fig. 3h).

**Collagen is plastically remodeled by the invading epithelium.** During the relaxation experiments we noticed that only a small fraction of the total deformation which accumulated during the growth process was released. These findings indicated that the observed deformations in the collagen network were predominantly plastic in nature (Fig. 3b).

As a consequence of these stable ECM deformations, we observed highly aligned collagen fibers and bundles in front of the invasive branches. Alongside the branches, fiber alignment was significantly reduced compared to the invasive tip, mimicking the initial anisotropic deformation field (Supplementary Fig. S4A, C). By contrast, the collagen network far away from organoids was fully isotropic with randomly oriented collagen fibers, indicating that collagen fiber alignment in front of elongating branches was generated by the expanding branches of the organoid (Supplementary Fig. S4B, C).

The plastic nature of the collagen network was further underscored by the observation that orientational order of the collagen network was kept in this alignment upon Cytochalasin D treatment (Fig. 3c, Supplementary Fig. S4C), thus maintaining deformation even after loss of tension. This plastic mechanical response of collagen can be observed clearly in cyclic shear rheology (Fig. 4a, Supplementary Fig. S5). Here, the stress–strain response depends on the number of initially applied shear cycles. Specifically, already after the first cycle (I) the slope of the stress–strain curve, which corresponds to the tangential modulus, diminishes significantly up to the previously maximal applied

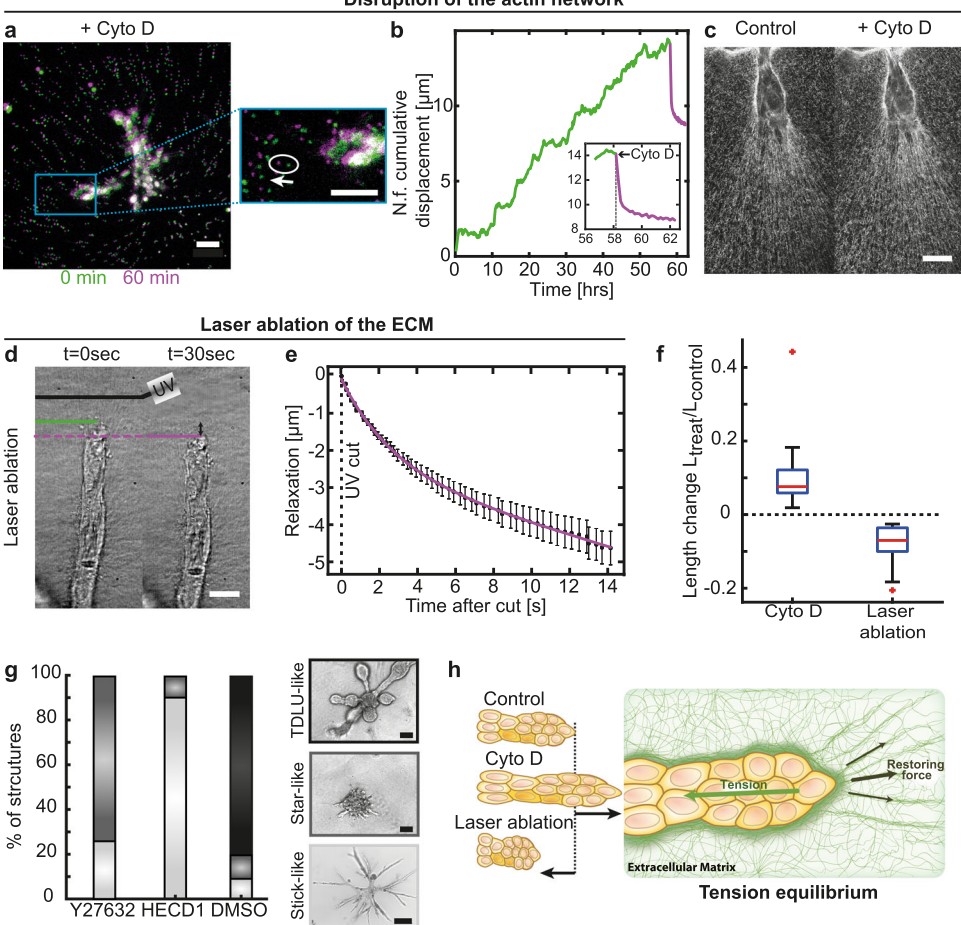

**Fig. 3 ECM deformations are enabled through a tension equilibrium. a** Upon treatment with Cytochalasin D organoid branches relax. Beads retract in the opposite direction of the initial deformation field (white arrow, $n = 31$ organoids). **b** Representative behavior of cumulative displacement in front of the branch during branch elongation (green) and relaxation upon treatment with Cytochalasin D (1 µg/ml, purple). **c** Fiber alignment in front of the branch is conserved after treatment with Cytochalasin D. **d** UV-cuts of the ECM in front of growing branches and following contraction of the branches towards the organoid body ($n = 121$ organoids). **e** Tracking the tip of a branch after a cut reveals a fast contraction towards the organoid body. Error bars, mean ± s.d. **f** After disruption of the actin network via Cytochalasin D the restoring forces of the ECM dominate, leading to branch elongation ($n = 31$ organoids). Contrary, after laser ablation, the dominating tension of the branch leads to branch shrinkage ($n = 21$ organoids). Box plots indicate median (red line), 25th, 75th percentile (blue box) and 5th and 95th percentile (whiskers) as well as outliers (single points). **g** Drug screening reveals loss of potential to grow TDLU-like structures upon Y-27632 ($n = 31$ organoids) and HECD1 ($n = 10$ organoids) treatment. **h** Tensile forces of the branches and the restoring forces of the aligned ECM lead to a tension equilibrium stabilizing the branches. Scale bars, 50 µm (**a**, inset **a**), 20 µm (**c**), 15 µm (**d**). Organoids were derived from three biologically independent donors (Supplementary Table S1). Source data are provided as a Source Data file.

strain, where a high stiffening is observed. This strain memory effect in polymeric materials is also known as Mullins softening and is based on structural changes inside collagen networks[29,30]. Each cycle leads to plastic fiber elongation[31] and the formation of weak crosslinks between approaching collagen fibers[32], which gradually changes the structural properties of the fibrillar matrix, similar to what has been demonstrated for crosslinked actin networks[33].

Live-cell imaging revealed that fiber alignment was induced by the collective and dynamic mechanical tension produced by the collective motion of cells within the elongating branches, mimicking a cyclic strain application (Supplementary Video 8). In addition, small range deformations were observable due to the invasion dynamics of the leading cells. These cells did not continuously attach to the same collagen fibers, but frequently changed their attachment sites within the collagen network, leading to only a localized but spatiotemporally inhomogeneous deformation field (Supplementary Fig. S4D). These short-ranging deformations occurred asynchronously to the large deformation

field which resulted from the collective cell motion and thereby caused small deviations from the correlation between collective motion and ECM deformation. Thus, the observed fiber alignment was a direct consequence of the observed contractile deformation field and captured its history due to the plastic properties of the collagen network.

By the use of fluorescently labeled collagen, we observed that this tension-induced collagen fiber alignment ultimately resulted in an enrichment of collagen along the branch axis leading to the formation of a continuous collagen cage (Fig. 4b). High-resolution microscopy revealed that this collagen cage had a porous structure with holes of an approximate size of 1 up to 3 µm and a thickness of up to 12 µm (Fig. 4c, Supplementary Video 9). Towards the tip of the branch the cage thinned out with larger holes at the invasion site, which are still significantly smaller than in the far field (Fig. 4d). Washing out the epithelial cell layer by addition of Triton X left an empty collagen cage behind, lining the borders of the vanished structure (Fig. 4e). Thus, the plastic deformation of the collagen generated a collagen

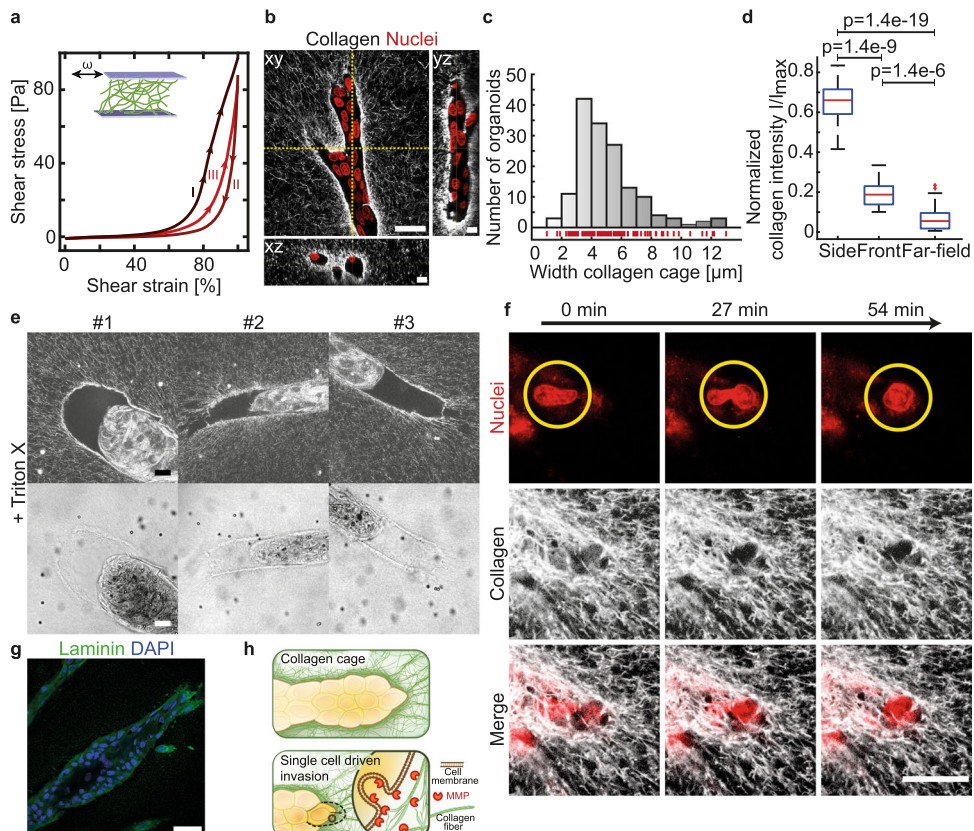

**Fig. 4 Collagen is plastically remodeled by the invading epithelium resulting in a stable collagen cage. a** Cyclic shear rheology of collagen shows plastic remodeling. **b** Collagen accumulation around growing organoids visualized by fluorescent collagen. A high-resolution reconstruction can be seen in Supplementary Video 9. **c** Distribution of the collagen cage width (*n* = 153 organoids). **d** Collagen intensity in dependency of position (Side *n* = 25 organoids, Front *n* = 25 organoids, Far field *n* = 25 organoids). Box plots indicate median (red line), 25th, 75th percentile (blue box) and 5th and 95th percentile (whiskers) as well as outliers (single points). **e** Organoids treated with Triton X. Upper panel: The collagen cage retains its structure even after the collapse of the branch. Lower panel: The cage is visible in the bright field image. **f** During ECM invasion leader cells squeeze through pores at the invasive front. **g** Immunostaining of laminin at day 11 of organoid growth showed localized expression at the cell-ECM interface. **h** Schematic representation of the collagen cage and the invasion of the tip. Scale bars, 30 μm (**b** xy), 10 μm (**b** xz and yz, **e**), 20 μm (**f**), 50 μm (**g**). Organoids were derived from three biologically independent donors (Supplementary Table S1). *P* values are from a two-tailed Mann–Whitney test and provided in Supplementary Table S5. Source data are provided as a Source Data file.

cage which encased the organoid and, once formed, remained mechanically stable even in the absence of cells. Together, these observations suggested that mechanical stability and anisotropy of the collagen cage guide the further elongation process of branch. While at the side the dense cage prevented further outgrowth[34], the front of the cage at the tip of the branch was porous and weak enough for individual cells to squeeze through (Fig. 4f, Supplementary Fig. S4E). Indeed, these results may also explain why formation of a new branch was observed exclusively through bifurcation, rather than side branching (Supplementary Fig. S6). In addition to accumulated collagen around the epithelial branches, we detected expression of laminin, a major component of the basement membrane (Fig. 4g).

To further determine this anisotropic accumulation of collagen around the branches we performed immunofluorescence staining of MMP9, a metalloproteinase previously described to have a dominant role in matrix remodeling in the mammary gland[35]. MMP9 staining revealed a localized expression at the tip of the invading branches (Supplementary Fig. S4F). In order to test the interplay of plastic deformation and local degradation of the matrix, we inhibited the activity of Metalloproteinases (MMPs) by the addition of Marimastat. At the beginning of the organoid establishment phase, addition of 10 μM Marimastat led to the formation of very short and thin branches, thus may prevent

efficient branching morphogenesis altogether (Supplementary Fig. S4G). The inhibition of MMP activity during the branch elongation phase induced an arrest of invasion by the branch tip cells into the collagen, although formation of filopodia was still observed (Supplementary Fig. S4H, I, blue line). Moreover, generation of tension by the branches was not significantly impeded, as shown by large strains in the collagen and further plastic deformation of the surrounding matrix (Supplementary Video 10). However, MMP-inhibition impaired elongation of the branch. Since proliferation of cells within the branches continued, the reduction in branch elongation observed upon MMP-inhibition resulted in increased cell density and a concomitant slowing of collective cell motility, eventually leading to thickening of the branches (Supplementary Fig. S4H, orange line). Based on these results, we concluded that the formation of the collagen cage resulted from a combination of mechanically induced accumulation of collagen in front of the branch and single-cell invasion of the tip cells with active degradation of the ECM (Fig. 4h, Supplementary Video 11).

## Discussion

When cultured in floating collagen type I gels, single primary human mammary cells of the basal lineage give rise to branched

multicellular organoids which invade the collagen matrix during branch elongation until branch outgrowth stops and alveoli-like buds form. Here, we observe collagen remodeling during branch elongation as a result of branch-internal collective cell migration characterized by a back-and-forth movement and a tension equilibrium between the branch and the surrounding matrix. We show that the plastic nature of collagen leads to the formation of a mechanically stable collagen cage surrounding the elongating branches, creating a spatial confinement for cellular movements within the branch and, thereby, guiding further elongation. Thus, formation of the collagen cage supports collective migration of cells within the elongating branches. This effect is reminiscent of the collective cell migration patterns observed in 2D model systems, when confinement is realized by spatially defined functionalization of the substrate[18,36]. In such 2D epithelial cell sheets, traction forces of individual cells are integrated over the whole epithelial layer via cell-cell junctions, ultimately leading to tension buildup spanning the full layer width[15]. In the elongating mammary gland organoid branches, we similarly observed strong cell-cell coupling through overarching alpha smooth muscle actin cables as well as adherens junctions evidenced by E-cadherin expression. These results, together with the observation that an E-cadherin-blocking antibody inhibited generation of tension and impaired branch elongation indicated a similar mechanism of tension generation within the branches. Together, our results suggest that confinement of cells within a mechanical stable collagen cage results in the ability of the cells to integrate their contractile force dipoles, leading to the exertion of large anisotropic tension with the ECM at the tip of the branch. This tension in turn was balanced by the mechanical properties of the ECM, which led to highly anisotropic plastic deformation together with the formation of highly aligned fiber bundles, as observed by us and others[20,21,32,37]. The aligned fiber bundles, in turn, exerted a restoring force to the branches, thereby generating tension equilibrium that ultimately enabled elongation of the branch. Similarly, in vivo, aligned collagen fibers have been shown to orient murine mammary epithelium outgrowth and it was proposed that such alignment is generated by stromal cells and the expanding epithelium itself[11,23]. Here, our model provides a detailed description of how epithelial self-organization results in collective tension buildup, which leads to the observed plastic fiber alignment that in turn guides further outgrowth.

Moreover, our data reveal that invasion into the ECM by the branches relies on local collagen degradation by the leading cells utilizing MMPs, comparable to single-cell invasion in dense matrices[38,39]. Specifically, we observe the expression of MMP9 at the tip of the branch. Interestingly, in the mammary gland, MMP9-expression has been described as a risk-factor predisposing for invasion of ductal carcinoma in situ-associated myoepithelial cells[40], supporting the invasive characteristics of branch outgrowth in our organoid assay. Moreover, beside its proteolytic activity, MMP9 has been shown to play a role in the regulation of collagen gel contraction by smooth-muscle-cells[41], suggesting an interplay between invasion and tension buildup similar to what we observe. In addition, we argue that it is the invasive migration of the leading cells that leads to the buildup of a mechanical encasing of the complete branch by a stable collagen cage, thereby reinforcing self-organization.

Analogous, in vivo, a thickened layer of collagen was found around murine mammary ducts where it was speculated to stabilize the duct and to provide mechanical constrain for further branch expansion[42–44], comparable to the role of the collagen cage in our assay. Due to the developing stable collagen cage at the sides of the branch, an invasion of cells into the ECM is prevented at this location. Similarly, it was shown that the invasion of single cells can be prevented by collagen

densification[34]. By contrast, at the front the thinner collagen cage is associated with collagen degradation by the leading cells, enabling branch elongation. Further, this mechanical constrain limits branch initiation to the invasive front via bifurcation, as it is observed for primary ducts during the expansion of the mammary epithelium during puberty[4,45]. However, in response to cycling ovarian hormones secondary side branches form along the existing ducts[4]. Similar to our model, in vivo this lateral branching is facilitated by the use of MMPs[46].

Around day 10 of organoid growth, we observed that more and more branches stopped invading into the matrix, rounded up and formed alveoli-like end buds. It can be speculated, that this final phase of organoid development is initiated by a lack of enough elastic restoring forces and local yielding of the remaining collagen matrix. In addition, at this stage of organoid growth, the collagen cage was observed to be fully closed at the invasive front preventing force transmission. End bud formation was accompanied by the deposition of laminin 1, an integral component of mammary gland basement membrane in vivo, as shown before[22].

In summary, our results reveal how branch elongation of human mammary gland organoids is governed by the external mechanical plasticity of the ECM. Importantly, E-cadherin mediated cell-cell coupling plays a crucial role throughout the whole self-organization process of human mammary gland organoids, similar to what has been observed in murine explants[47]. It will be of interest to determine other biochemical signaling pathways involved in the observed oscillatory contractile behavior such as cycles of MLC (myosin-light-chain) phosphorylation and dephosphorylation via Rho/ROCK and FAK/Rac signaling[48]. Due to the universal role of branching morphogenesis in organogenesis, the dynamic mechanical tension equilibrium identified in this study paves the way to further address mechanical self-organizing principles in morphogenesis.

## Methods

**Isolation of human mammary cells from reduction mammoplasty samples.** Human breast tissue from healthy women undergoing esthetic reduction mammoplasties at the Nymphenburg Clinic for Plastic and Aesthetic surgery was obtained in accordance with the regulations of the ethics committee of the Ludwig-Maximillians University, Munich, Germany (proposal 397-12) which includes patient education and consent and irreversible anonymization of tissue samples. Information concerning age and parity of donors is summarized in Supplementary Table S1. Breast tissue was directly taken from the operating room and immediately processed. Breast tissue was cut using scalpels and minced into 1 mm$^3$ pieces. The minced tissue was collected and digested sequentially using 300 U/mL collagenase I (Sigma) together with 100 U/mL hyaluronidase (Sigma), 0,15% Trypsin-EDTA (Life Technologies) and 5 mg/mL dispase (Stem Cell Technologies), and then cryopreserved. After thawing and before further processing, the cell suspension was filtered through a 40 µm strainer, to remove larger cell aggregates and residual tissue fragments.

**Flow cytometry and fluorescence-activated cell sorting (FACS).** Single-cell suspension of primary mammary cells were stained with the following antibodies: CD31-PB, CD45-V450, CD49f-PE, EpCAM-FITC and CD10-APC (Supplementary Tables S2–S4). 7AAD was added to the suspension for dead cell exclusion. Luminal progenitors (LP) and CD10+ basal cells (B+) were sorted as previously described (Fig. 1a)[22]. A subsequent re-analysis was performed to ensure the purity of the sort. FlowJo V10 software was used for post-analysis.

**Organoid preparation.** Freshly isolated human mammary gland epithelial cells from healthy women undergoing reduction mammoplasty were embedded in collagen gels (collagen type I from rat tail, Cornings) with a final collagen concentration of 1.3 mg/ml. For specific experiments pure collagen was mixed with labeled collagen in a ratio of 20:1. From day 1 to 5 cells were cultivated in mammary epithelial growth medium (PromoCell MECGM) enriched with 3 µM Y-27632 (Biomol), 10 µM Forskolin (Biomol) and 0.5% FBS. From day 5 to day 14 the media was changed to MECGM mixed with 10 µM Forskolin. Experiments were repeated for organoids prepared from at least 3 different donors. In total, 5 different donors were used for organoid preparation (Supplementary Table S1).

**Collagen labeling**. Collagen was fluorescently labeled with Atto 488 (Merck) according to a protocol based on a previously published protocol[49]. Therefore, collagen was dialyzed at 4 °C to reach pH 7. Subsequent, collagen was conjugated with Atto 488 by incubating it overnight at 4 °C. Further dialyze was performed for 8 h to remove non-bound dye, followed by an additional dialyze overnight using acid to prevent unwanted polymerization. Finally, it was stored at 4 °C.

**Live-cell imaging**. Live-cell imaging was done using a Leica SP8 lightning confocal microscope and data was collected with the Leica Application Suite X v. 3.57. The microscope was equipped with an ibidi gas incubation system for CO2 and O2. Organoids were labeled with sirDNA (10 μM, Spirochrome AG) 3 h before measurement. Organoid age and observation time was set according to the planned experiment time between each image was kept at 10 min. For nuclei visualization samples were excited with 633 nm and detected at an emission maximum around 674 nm, detected with a HCX PL APO 10x/0.40 CS dry objective. Collagen fibers were either visualized by illuminating the samples with 488 nm and detecting the reflected signal or by imaging fluorescently labeled collagen using a HC PL APO 40x/1.10 water immersion objective. High-resolution images of the collagen cage were taken by using an implemented adaptive deconvolution algorithm and the usage of a HC PL APO 63x/1.40 oil immersion objective.

**Inhibitor treatment**. To analyze the influence of specific pathways and processes during branching morphogenesis different inhibitors were used. To inhibit actin polymerization organoids were treated with 1 μg/ml Cytochalasin D (Merck). organoids were analyzed 4 h after treatment. To inhibit collagen degradation 10 μM Marimastat (Merck) was added to the organoid culture. According experiments were started directly after treatment. Inhibition of Rho kinase was achieved by using 10 μM Y-27632 (Abcam). Blocking of E-cadherin was performed by addition of an E-chadherin blocking-antibody HECD1 (Abcam) in a dilution of 1:50. Specifically, HECD1 was added continuously from day 5 to day 14.

**Data analysis**. Images were processed using Fiji and Matlab. Drift correction was performed using a self-written Matlab code based on maximizing the correlation value between consecutive images by shifting images stepwise in x-y-z direction. For bead detection images were first masked by an intensity threshold. Subsequent bead position is defined by taking the center of an interpolated intensity grid. Beads touching the boundary of the image and beads below a minimum distance to each other are removed to prevent wrong trajectories. Finally, tracks are calculated by matching coordinates in three consecutive images. To calculate the cumulative displacement, the individual displacement of a bead in the near field was summed up and afterwards the mean was calculated by averaging of beads in the same area. To calculate the displacement in dependency of the distance to the organoid, we segmented the ECM in front of the branches in stripes with a width of 100 μm. In those stripes we averaged the displacement of the beads and calculated the average displacement. The internal flow field was calculated via optical flow using the in Matlab implemented Farneback algorithm. We calculated the distribution of fiber angles, fitted a gaussian to the according histogram and calculated the full width half maximum (FWHM). We defined the degree of alignment $d$ as:

$$d = \frac{(FWHM_{max} - FWHM)}{FWHM_{max}} 1$$

Hereby, $FWHM_{max}$ refers to the maximal FWHM measured in all conditions.

**Immunofluorescence**. Collagen gels were fixed with 4% paraformaldehyde for 15 min. For immunofluorescence stainings, cells were permeabilized with 0.2% Triton X-100 and blocked with 10% goat or donkey serum in 0.1% BSA. Samples were incubated with primary and secondary antibodies diluted in 0.1%BSA. Antibodies and dyes used for stainings are listed in Tables 2, 3 and 4. DAPI was used to visualize the cell nuclei.

**Laser ablation**. Laser ablation of collagen was performed with a custom-made nano-dissection setup based on the one described previously[50]. Our setup includes a spinning-disc unit (CSU-X1, Yokogawa) equipped with an Andor NEO sCmos camera, a 638 nm and a 488 nm laser (Cobalt) for fluorescence confocal imaging, as well as a HAMAMATSU Orca Flash 4 for bright field imaging along a separate optical path. Cuts were done by shining a pulsed UV-laser with emission of 355 nm, 400 psec pulse duration, 72 kW peak power and 25 mW average power through a HC PL APO 40x/1.10 water objective (Leica microsystems) with a working distance of 0.65 mm. Movies were taken at frame rates between 120 fps and 200 fps. For analysis movies were processed with Fiji. The tip cell was tracked manually to measure the relaxation after the cut.

**Rheology**. Rheological experiments were performed using an Anton Paar Physica MCR 301 plate rheometer. Polymerization of the hybrid gels was monitored using oscillatory shear experiments with deformations within the linear regime. Thereby, the storage and the loss modulus were analyzed over time to ensure the complete gelation of the gels. Afterwards, frequency sweeps with gradually increasing amplitude were performed to analyze the plastic behavior of pure collagen networks (Supplementary Fig. S5).

**LifeAct-plasmid, virus production and infection of target cells**. HEK293T high performance cells (ATCC) were transfected with pLenti.PGK.LifeAct-GFP.W (Addgene plasmid 51010). The cell-free supernatant was collected after 48 h and the virus titer was enriched via ultracentrifugation. Primary human basal mammary epithelial cells were transduced and after 24 h, cells were trypsinized and seeded into floating collagen gels.

**Statistics and reproducibility**. Matlab R2019 was used for statistical analyses. P values are from a two-tailed Mann–Whitney test and provided in Supplementary Table S5. All experiments were repeated at least for three independent donors unless otherwise indicated, exact number of observed phenomena are reported in the figure legends.

**Reporting summary**. Further information on research design is available in the Nature Research Reporting Summary linked to this article.

## Data availability
Microscopy data that support the findings of this study have been deposited in Zenodo with the identifier https://doi.org/10.5281/zenodo.4590475. All other relevant data supporting the key findings of this study are available within the article and its Supplementary Information files or from the corresponding author upon reasonable request. Source data are provided with this paper.

## Code availability
Matlab codes for bead tracking and optical flow analysis can be found on Zenodo with the identifier https://doi.org/10.5281/zenodo.4590475.

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

## Acknowledgements

We gratefully acknowledge the financial support of the European Research Council (ERC) under the European Union's Horizon 2020 research and innovation programme (grant agreement No. 810104 - PoInt) and the Deutsche Forschungsgemeinschaft (DFG, German Research Foundation) – Project-ID 201269156 – SFB 1032. We thank Christian Gabka from the Nymphenburg Clinic for Plastic and Aesthetic Surgery, Munich 80637, Germany for providing primary human mammary gland tissue.

## Author contributions

A.R.B, C.S, B.B. and L.K.E conceived the experiment. B.B. conducted live-cell imaging and analyzed the deformation fields. L.K.E. performed immunofluorescence and drug screening. P.F. carried out the shear rheology measurements. M.K.R. applied the laser ablation experiments. F.P.H. analyzed the structure of the collagen networks. All authors contributed to the interpretation of the data and to the writing of the manuscript.

## Funding

## Competing interests

The authors declare no competing interests.
