## [Peer Review File · Nature Communications]

REVIEWER COMMENTS

Reviewer #1 (Remarks to the Author):

Manuscript: Mechanical plasticity of the ECM directs branch elongation in human mammary gland organoids

General comments: This manuscript uses human mammary organoids that are cultured in floating collagen gels to investigate how plasticity of the collagen matrix affects branch elongation. Consistent with previous studies (doi: 10.1016/j.cub.2013.03.032) of mammary organoids, the authors find that branch elongation deforms the collagen matrix and aligns the collagen fibers in front of the branch. They also report that cells rearrange dynamically in elongating branches, which is consistent with previous observations of mammary organoids (<https://doi.org/10.1038/s41567-019-0680-8>). The authors find that tension created by the collective motion of cells within branches leads to tension built up in the surrounding collagen matrix that leads to plastic deformation. Furthermore, the authors find that plastic remodeling of the type I collagen matrix generates a mechanically stable collagen cage surrounding branching mammary organoids. They conclude that the collagen cage directs tension generation, branch outgrowth, and plastic deformation of the collagen matrix. The data in the manuscript supports most of the authors' claims (see comments below); however, the novelty of the current submission is unclear. Additional work is needed to increase the clarity of the manuscript and place the results in the context of previous mammary organoid research and in vivo observations of mammary gland development.

Specific concerns:

1. The authors have missed some important references in this manuscript:
 - a. When discussing how radial intercalation contributed to branch elongation (page 3), the authors should include the following paper: Neumann et al. *Dev Cell* (2018); 45(1):67-82.e6. doi: 10.1016/j.devcel.2018.03.011.
 - b. When discussing the mechanisms that guide collective outgrowth of epithelial structures, the authors should include the following paper: Piotrowski-Daspit et al. *Biophys J* (2017); 113(3):702-713. doi: 10.1016/j.bpj.2017.06.046.
2. The plot in Figure 1G is confusing. The clarity of this figure could be improved if each line in the plot corresponded to an individual bead at a certain distance from the organoid. Alternately, the authors could explain the meaning of the schematic at the bottom of the plot in greater detail in the figure legend. The differences between lines in the plot would also be clearer if both the line color and line style were varied.
3. On page 3 of the manuscript, the claim that "those exchanges occurred independent of branch length and width during the whole process of branch elongation" is not supported by data in the manuscript. To support this claim, the authors need to quantify the number of exchanges as a function of branch length and width and show that there is no statistically significant difference.
4. On page 3 of the manuscript, the claim that "no directed cell movements and ECM deformations were observed perpendicular to the long axis of the branches" is not supported by data in the manuscript. To demonstrate this, the authors need to quantify displacement of beads and cells along the axis perpendicular to the long axis of the branches. Also, how is the data in Figure 2D consistent with this claim? It shows high cell velocities in the y direction for a branch that is extending in the x

direction.

5. On page 3 of the manuscript, the authors state that “strikingly, the direction of the net movement of cells was synchronous with the deformation field within the ECM in front of branch tips”. Why is this striking? What else would you expect to observe? Additional experimental context is needed to further support the above statement.

6. On page 4 of the manuscript, the authors state that “Here, the alignment of fibers showed to decrease with increasing angle to the branch axis (Figure S3A)”. This claim is not well-supported by the data. The authors need to quantify the orientation of fibers as a function of angle to the branch axis as was done in Figure 1E for bead displacement. Similarly, the authors should quantify collagen orientation in Figure S3B as a function of distance from the organoid to show how far you need to be from the organoid to see an isotropic network of collagen fibers.

7. On page 4 of the manuscript, the authors state “the orientational order of the collagen network was kept in this alignment upon Cytochalasin D treatment (Fig. 3C)”. How long after Cytochalasin D treatment was the image in Figure 3C acquired? Also, to fully support this claim, the authors need to quantify collagen fiber alignment in front of the branch and show that there is no statistically significant difference after Cytochalasin D treatment.

8. It is difficult to see nuclear deformation in Figure 4F. The clarity of this figure could be improved with higher magnification images or by showing the collagen and nuclei staining channels separately.

9. What is the purpose of Figure S4? It is not referenced anywhere in the text apart from the methods section. It also contains analysis for Matrigel, which was not included in the data in the manuscript.

10. The methods section of the manuscript does not contain enough information for an independent researcher to reproduce the experiments. For example, the immunofluorescence section should include antibody dilutions used. In addition, the methods section should include information about the concentration of Cytochalasin D (only listed in description of video S5), marimastat (listed at various points in the manuscript but not in the method section), and Y27632.

11. The article could be improved by discussing the relevance of the findings to in vivo mammary epithelial branch elongation. Is the concept of a collagen cage relevant for in vivo branching morphogenesis? Is the concept of ECM plasticity directing branch elongation consistent with the pattern of the mammary epithelium in vivo?

Additional comments that will help improve clarity and overall quality of the article:

1. Please consider avoiding the use of red and green in schematics (e.g. Figure S1D). These figures are difficult to see for anyone who is red-green colorblind. Red can be replaced with magenta or gray to avoid this problem. Here is the link for a helpful article: <https://www.ascb.org/science-news/how-to-make-scientific-figures-accessible-to-readers-with-color-blindness/>

2. Error bars should be added to Figure 1E. Otherwise, it is unclear how reproducible the observed decrease in displacement is.

3. The phrase “Analogous, also branch elongation” on pg 3 of the manuscript is unclear and should be reworded.

4. The sentence “In order to identify, the mechanism...” on page 3 does not need a comma.

5. The labels for LifeAct and Nuclei in Figure 2A and D, respectively, are hard to read. Please consider reformatting.

6. In Figure 2D and E, it is hard to compare the data because the velocity is reported as arbitrary units in Figure 2D and as $\mu\text{m}/\text{min}$ in Figure 2E. Consider using consistent units for the entire figure to enable a comparison.

7. In Figure S2D the magenta channel seems to show nuclei and the position of the beads. Adjust the

label accordingly.

8. In Figure 3H, the authors refer to a “thin stick-like structure” while on page 4 of the text they refer to “very thin and spindly branches structures”. Please be consistent with the description of the organoid morphology. Otherwise, it will confuse some readers.

9. The order of figure panels in Figure S3E-H does not match the text. Also consider showing expression of MMP9 first, before showing the results of blocking MMPs during the establishment phase and the elongation phase to avoid confusion.

10. On page 12 of the manuscript the authors write “according to the planned experiment, time”. The comma can be removed from this sentence.

11. On page 13 of the manuscript, the authors write transduced and after 24hours”. Space missing between 24 and hours.

12. The scale bar value for panel B is missing the figure legend of Figure 4.

13. The scale bar value for panel D is missing from the legend for Figure S1.

14. Y-27632 is referred to as Y27632 or Y-27632 throughout the paper. Please be consistent with the formatting.

Reviewer #2 (Remarks to the Author):

In this clearly written manuscript, Buchmann et al. use a 3D model of mammary epithelial branching in collagen to describe the tensile forces at play in the process of branching morphogenesis. Using live confocal imaging, the authors analyze the temporal and directional characteristics of cellular movement and ECM deformations and identify periodic collective movement of cells, leading to stable ECM remodeling which ultimately directs branch elongation. The authors nicely demonstrate the existence of a tension equilibrium between the elongating branch and the extracellular matrix, and describe the formation of a stable collagen cage that encapsulates the epithelial branch.

Overall, while this work uses a compelling model to characterize tensile forces enacted on the ECM by epithelial cells and vice versa, most of the observations described are not novel. In addition, only a superficial description is made on observations that are further investigated, and no mechanism is provided.

Data presented here that is not novel:

1. The need for Rho-kinase for branching morphogenesis (described in ref 21)
2. The permanent nature of the ECM changes as a result of cell movements (described in ref 19 and by others)
3. The role of collective cell migration in remodeling the ECM through tensile forces (described in ref 9 and others)

Findings presented here that were not further investigated:

1. The periodic nature of extension and retraction of cell movements that correlate with ECM deformation: this was previously observed in ref 9, but was not elaborated on there, either. Here, it was more thoroughly described, but there was no attempt to test the necessity of this periodic nature for ECM remodeling or branching morphogenesis, or to find the underlying mechanism behind the periodic collective cell movement.
2. Tip cell overtaking: this is a recorded phenomenon in endothelial branching (Boas and Merks, BMC

Syst Biol 2015), but its significance or underlying mechanism are not known. It is clearly demonstrated and comprehensively described here, but not investigated or discussed further. It would be interesting to know if it is entirely stochastic and whether it plays a role in the branching/elongation process.

3. The formation of a stable collagen cage: the authors describe the collagen cage, but only assume the nature of its formation through tensile forces. Experimental interventions are needed in order to understand:

i) the role of collagen secreted by the cells vs. pre-existing matrix collagen in the formation of the cage (laminin is shown to be secreted by the cells, was collagen secreted too?)

ii) the role of the cage in preventing or directing side branching as well as directional elongation through the branch tip (It would be interesting to manipulate areas of the cage to see how it affects branching and elongation)

iii) other roles the cage may play in mediating epithelial/ECM interactions.

Importantly, as this finding is among the highlights of this work, it seems essential to demonstrate how a collagen cage formed in vitro relates to branching morphogenesis in vivo: is there evidence of an equivalent matrix cage in the breast? How does an in vitro collagen cage correspond with in vivo basement membrane?

Specific comments:

- Most of the information presented in S1A-D was previously shown in Linnemann, Development 2015.
- Fig. 1G: it is difficult to tell the two brown hues apart in the graph.

Reviewer #3 (Remarks to the Author):

The manuscript titled “Mechanical plasticity of the ECM directs branch elongation in human mammary gland organoids” presents an interesting characterization of how a balance of cell and ECM tension, along with cyclic collective cell motility, drives large scale rearrangement of the ECM to form a collagen cage and promotes ductal branch elongation and invasion. The strengths of the study include a new and relevant experimental model (primary human mammary organoids derived from single basal cells in detached collagen gels), quantitative imaging of the organoids and surrounding ECM, including timelapse microscopy. The finding that cyclic changes in the direction of collective motility (periodic out/in behavior) can drive large scale changes in ECM alignment is particularly exciting and sheds new light into the dynamical mechanisms by which organoids remodel the ECM and ducts elongate. The study can benefit from additional characterization of the organoid system and how cellular tension and cell cortex coupling (adhesion-mediated) affects the cyclic nature of cell movement. Further discussion on how this body of work fits the existing literature on mammary gland morphogenesis (both in vivo and in vitro), as well as model system of collagen remodeling in 3D culture, will be helpful.

Main comments:

*Because this is a relatively new model system, it’s important to report on how efficient each stage in the organoid morphogenesis is, and provide more details on how specific organoids are selected

for analysis. Important points to describe include: What is the efficiency with which single cells establish the stage 1 organoids? What is the frequency of transition between stage I/II; stage II/III? What is the frequency with which GATA3(+) organoids form from stage I organoids? Is the time point of GATA3(+) cell emergence stereotyped? If so, what is the structural event that correlates with GATA3 cell emergence? Without this information, it would be near impossible for another lab to evaluate whether they were successfully reproducing the results. Further, it would make building on the work challenging.

*The fact that basal/myoepithelial cells express both E-cad and P-cad -- the latter expressed at higher levels -- might suggest that the role of E-cad in these structures is more complex than simply coupling cell contractility or cortices. How does cell adhesion/coupling of cell cortices affect coordinated cell movement? What is the role of P-cadherin in these movements? For example, does the E-cad blocking antibody (or P-cad) reduce cell coupling which reduces the coordination between cells and impairs branch elongation.

*Aside from lacking alveologenesis, which does not seem to be the focus of the rest of the manuscript, how different are the anti-E-cadherin-treated organoids during the branching morphogenesis stage? This seems like an important experiment for demonstrating the importance of collective tension, so it would support the argument if you included quantification of branching, cell motility, long-range ECM deformation, or ECM alignment for this condition. For example, is collective tension dispensable or necessary for the formation of a collagen cage or aligned fibers at the branch tip?

Minor comments:

*The discussion of “small range deformations ... due to the invasion dynamics of the leading cells ... asynchronous to the large deformation field” seems to be missing a transition to “Thus, the observed fiber alignment was a direct consequence of the observed contractile deformation field”. The purpose of the paragraph seems to be that, despite short-range deformations based on the invasion dynamics of the leader cells, the overall fiber alignment is caused by the collective motion of cells. If so, then such an argument would be well-supported by a description of the collagen fiber alignment associated with treatments that do not affect the invasion behavior but do affect collective motion, like in the anti-E-cadherin experiment described in a previous section.

*In discussion, “at the front the thinner collagen cage permits collagen degradation by the leading cells” is not quite congruent with the results, which instead point to “at the front the thinner collagen cage is associated with collagen degradation by the leading cells”.

*The manuscript is a detailed characterization of invasive branching morphogenesis of mammary organoids in a collagen gel. The introduction references branching morphogenesis in development, and the discussion would be more impactful if it could compare the observations in this work with the existing literature on the process of mammary branching morphogenesis, both observed in vivo or in other mammary organoids.

Figure 2

2A: Caption could be more descriptive, perhaps highlighting the actin-based protrusions or filopodia/invadopodia visible at the top edge.

2B: For clarity, it may be helpful to rotate the images so that they have the same orientation as the

branch in 2A (tip pointing up).

2C: Title is somewhat confusing because “events” are not “branches” (y axis). Could be rephrased as “Branches with exchanges within 24 hrs”

2D: The separation of velocity into two separate graphs for the x and y component is somewhat confusing and may detract from the purpose of the panels (showing clusters of collective motion in the same direction). Could this be done in a different way that is more clear, for example, by color-mapping angular direction or by plotting a vector field?

2E and 2G have the same time scale, but seem to suggest different takeaways about the dynamics of the system. In 2E, it looks like there is a 200-min period of inward migration, followed by 100+ minutes of outward migration. In 2G and H, it looks like periods of inward and outward migration switch every twenty minutes or so.

Figure 3.

3H: Could the legend labels (TLDU-like, Star-like, Thin stick-like) be vertically distributed to align with the matching inset images? This makes it more clear that these images are intended to complement the legend.

Some figures may be inaccessible to people with colorblindness, such as Figure 2 and Figure S1, which use red and green.

Minor comments (text):

“long ranging deformation” should be “long-range deformation”

“We observed a high expression” should be “We observed high expression”

“A phalloidin staining” should be “Phalloidin staining”

“Accordingly, the ECM was relaxing” should be “Accordingly, the ECM relaxed”

“Thus, the tension is built up by the whole branch and is equilibrated by the surrounding ECM” seems to belong after the following paragraph, after discussing the anti-E-cadherin antibody.

“collective tension built-up” should be “collective tension buildup” here and in other places where “built-up” is used as a noun rather than an adjective

“Here, the alignment of fibers showed to decrease with increasing angle” should be “Here, the alignment of fibers decreased with increasing angle”

“Each cycle, leads to plastic fiber elongation” should not have the comma there

“Leading to only a localized but spatiotemporal inhomogeneous deformation field” might be intended to be “leading to not only a localized but also a spatiotemporally inhomogeneous deformation field”

“We observed that this tension induced collagen fiber alignment” should be “We observed that this tension-induced collagen fiber alignment”

“While at the side the dense cage prevents a further outgrowth” should be “While at the sides the dense cage prevents further outgrowth”

We thank all three referees for their constructive suggestions and comments, which we are happy to address in the current version of the manuscript. To this end we performed a series of new experiments and analysis. By the help of the referees, we are convinced to have improved the manuscript significantly.

REVIEWER COMMENTS

Reviewer #1 (Remarks to the Author):

Manuscript: Mechanical plasticity of the ECM directs branch elongation in human mammary gland organoids

General comments: This manuscript uses human mammary organoids that are cultured in floating collagen gels to investigate how plasticity of the collagen matrix affects branch elongation. Consistent with previous studies (doi: 10.1016/j.cub.2013.03.032) of mammary organoids, the authors find that branch elongation deforms the collagen matrix and aligns the collagen fibers in front of the branch. They also report that cells rearrange dynamically in elongating branches, which is consistent with previous observations of mammary organoids (<https://doi.org/10.1038/s41567-019-0680-8>). The authors find that tension created by the collective motion of cells within branches leads to tension built up in the surrounding collagen matrix that leads to plastic deformation. Furthermore, the authors find that plastic remodeling of the type I collagen matrix generates a mechanically stable collagen cage surrounding branching mammary organoids. They conclude that the collagen cage directs tension generation, branch outgrowth, and plastic deformation of the collagen matrix. The data in the manuscript supports most of the authors' claims (see comments below); however, the novelty of the current submission is unclear. Additional work is needed to increase the clarity of the manuscript and place the results in the context of previous mammary organoid research and in vivo observations of mammary gland development.

We thank the reviewer for the highly constructive, thorough and supporting review that allowed us to strengthen and clarify many of our observations and provide a more concise, stronger manuscript. In addition to the new experimental data and analysis we took special emphasis in improving clarity of the manuscript and emphasizing better the novelty of the findings.

While it has been merely observed that the cells interact with the collagen matrix, the present work now is the first to unambiguously demonstrate that the interaction between cells and matrix leads to the formation of a stable collagen cage, which in turn is

prerequisite for the observed collective cell migration resulting ultimately in the plastic deformations of the ECM. Our results summarize into a general mechanical tension model of the self-organizing process which sets the base for addressing even more complex morphogenetic processes found in organogenesis. *In vivo*, such collagen accumulations have been observed around the murine mammary epithelium, where it was speculated to stabilize the duct and constrain its expansion. As suggested, we highlighted the *in vivo* relevance in the discussion. We like to point out that our assay is a unique model system for analyzing branch elongation of primary donor specific human tissue, as the arising organoids resemble the TDLU-like morphology and bilayered architecture of the human epithelium *in vivo*.

Specific concerns:

1. The authors have missed some important references in this manuscript:

a. When discussing how radial intercalation contributed to branch elongation (page 3), the authors should include the following paper: Neumann et al. *Dev Cell* (2018); 45(1):67-82.e6. doi: 10.1016/j.devcel.2018.03.011.

b. When discussing the mechanisms that guide collective outgrowth of epithelial structures, the authors should include the following paper: Piotrowski-Daspit et al. *Biophys J* (2017); 113(3):702-713. doi: 10.1016/j.bpj.2017.06.046.

We thank the reviewer for pointing out these publications and added them to the manuscript.

2. The plot in Figure 1G is confusing. The clarity of this figure could be improved if each line in the plot corresponded to an individual bead at a certain distance from the organoid. Alternately, the authors could explain the meaning of the schematic at the bottom of the plot in greater detail in the figure legend. The differences between lines in the plot would also be clearer if both the line color and line style were varied.

We changed the figure according to the reviewer's suggestion and specified the caption of the figure. When analysing the displacement in dependency of the distance to the organoids we segmented the ECM in stripes with a width of 100 μm and averaged over the beads inside these stripes. To further clarify this, we added a paragraph to the Materials and Methods section. In our opinion, comparing single beads might be misleading due to the inhomogeneity of the deformation field. However, the deformation of single beads can be also seen in Figure 1D.

Changes were made: Fig. 1G and caption, Materials and Methods

3. On page 3 of the manuscript, the claim that “those exchanges occurred independent of branch length and width during the whole process of branch elongation” is not supported by data in the manuscript. To support this claim, the authors need to quantify the number of exchanges as a function of branch length and width and show that there is no statistically significant difference.

Following the suggestion of the reviewer, we now analysed branch length and width for branches with and without a tip exchange. Interestingly, we observed that a tip exchange is slightly more probable in shorter and narrower branches. We added this observation in the manuscript and extended the supplements with the related analysis (Fig. S2).

Changes were made: Manuscript pg. 3; Additional data: Fig. S2 A-D

4. On page 3 of the manuscript, the claim that “no directed cell movements and ECM deformations were observed perpendicular to the long axis of the branches” is not supported by data in the manuscript. To demonstrate this, the authors need to quantify displacement of beads and cells along the axis perpendicular to the long axis of the branches. Also, how is the data in Figure 2D consistent with this claim? It shows high cell velocities in the y direction for a branch that is extending in the x direction.

We followed the suggestion of the referee and calculated the bead displacement field in front and at the sides of developing branches and also changed the depiction of the velocity field within the organoids in Figure 2D. In addition, we plotted the temporal orthogonal velocity distribution of a branch in Figure S2E.

In all cases of extending branches, we observed that the deformation field at the sides is negligible compared to the deformation at the front. We computed that the deformation in the front is up to 15 times higher than at its sides. Additionally, we observed the periodic nature of bead contraction and relaxation only at the front, while at the sides of the branch, we did not see significant bead displacement. We added a representative analysis in Figure S1E.

Addressing the concerns of the reviewer, we revised Figure 2D. Instead of plotting velocities in x- and y-direction we now plotted the velocities in parallel and orthogonal direction to the branch axis. Thereby, we increased the comparability between the two different velocity components. However, we still observed cells migrating in orthogonal direction. This is either due to local cell rearrangements with their neighbours or due to the described uncoordinated migration. Yet, these velocities are up to five times slower than the velocities in parallel direction.

Changes were made: Fig. 2D; Additional data: Fig. S1E, Fig. S2E

5. On page 3 of the manuscript, the authors state that “strikingly, the direction of the net movement of cells was synchronous with the deformation field within the ECM in front of branch tips”. Why is this striking? What else would you expect to observe? Additional experimental context is needed to further support the above statement.

Indeed, this observation is striking as previous studies in other systems have shown that stalk cells are simply dragged behind the tip cells invading the ECM without any noticeable deformation. Here, cell migration within the whole branch causes the deformation. This is an example of invasion coupled to collective back-and-forth migration in order to exert mechanical force.

Compared to other invasive branching modes, where the stalk cells mainly support branch elongation via intercalations, we observe that the follower cells are not passively dragged behind the leader cells but migrate collectively with them. This collective back-and-forth migration results in a synchronized deformation field in the ECM. In total we analyzed the correlation between internal cell migration and external deformation field for n=11 organoids of three different donors, as can be seen in Figure 2G. Specifically, we calculated the optical flow within time series of extending branches. We separated the images into two regions, one region which contained the nuclei signal of the branch and

another which contained the ECM in the near field of the branch. Afterwards, we calculated the mean of the optical flow of each region to compare the direction and magnitude of the flow. In all analysis we saw a synchronization of the internal migration and the external deformation field.

However, due to style recommendations, we followed the reviewer's suggestion, and removed the term 'Strikingly' from the text.

Changes were made: Manuscript pg. 3

6. On page 4 of the manuscript, the authors state that “Here, the alignment of fibers showed to decrease with increasing angle to the branch axis (Figure S3A)”. This claim is not well-supported by the data. The authors need to quantify the orientation of fibers as a function of angle to the branch axis as was done in Figure 1E for bead displacement. Similarly, the authors should quantify collagen orientation in Figure S3B as a function of distance from the organoid to show how far you need to be from the organoid to see an isotropic network of collagen fibers.

We thank the reviewer for this advice. We calculated a spatial fibre alignment field, added the results to Fig. S4C and changed the text accordingly. We now compare the bead displacement and the collagen alignment in front and at the sides of developing branches. Here, we see that in regions with higher bead displacement the collagen alignment is significantly higher than in regions with very low bead displacements. Specifically, for the calculation of the spatial fibre alignment field we defined three different regions: the first directly in front of the branch, the second at the side of the branch and the third in an area where we did not see any alignment. All regions had a defined size of 100x100 µm. For each measurement we fitted a gaussian function to the histogram of fibre angles and calculated the according full width half maximum (FWHM). We defined the degree of alignment as following:

$$\text{degree of alignment} = \frac{(FWHM_{max} - FWHM)}{FWHM_{max}}$$

where $FWHM_{max}$ is the maximal calculated FWHM. By this we could show that there is significantly less alignment at the sides of the branches than at the front of the invading branches. Of note, we detected significant alignment at the sides of the branches. Since we do not see significant bead displacements and collagen deformation during live cell imaging, we conclude that the collagen alignment detected here is the result of the prior invasion process by the branch tip, leaving an imprint in the collagen matrix as branch elongation progresses.

Changes were made: Manuscript pg. 4, Materials and Methods; Additional data: Fig. S4C

7. On page 4 of the manuscript, the authors state “the orientational order of the collagen network was kept in this alignment upon Cytochalasin D treatment (Fig. 3C)”. How long after Cytochalasin D treatment was the image in Figure 3C acquired? Also, to fully support this claim, the authors need to quantify collagen fiber alignment in front of the branch and show that there is no statistically significant difference after Cytochalasin D treatment.

We thank the reviewer for this valuable suggestion. To better support our claim, we further quantified the fibre alignment after treatment with Cytochalasin D and plotted the results

in Fig. S4C. The aim of this experiment was to highlight the plastic matrix remodelling of the invading branches. In order to do so, we inhibited tension built-up via addition of Cytochalasin D and imaged the collagen network after no elastic relaxations were observable anymore. Specifically, organoids were imaged 4 hours after treatment with Cytochalasin D, as we did not see any relaxation of either the ECM or the branches anymore (compare Fig. 3B). Afterwards, we calculated the degree of alignment in front of the branches as described above. Here, the degree of alignment showed no significant difference to the alignment field of control organoids and thereby underlines the plastic collagen fiber alignment field.

Additional data: Fig. S4C

8. It is difficult to see nuclear deformation in Figure 4F. The clarity of this figure could be improved with higher magnification images or by showing the collagen and nuclei staining channels separately.

According to the reviewer's suggestion we chose a higher magnification image for Figure 4F and show plotting of the nuclei and collagen staining in separated figures as well as a merged figure.

Changes were made: Fig. 4F

9. What is the purpose of Figure S4? It is not referenced anywhere in the text apart from the methods section. It also contains analysis for Matrigel, which was not included in the data in the manuscript.

Due to changes in the supplementary figures, Figure S4 changed to Figure S5. Now, Figure S5 displays the extended data of the rheological characterization of a pure collagen network, which is partly depicted in Figure 4A. Explicitly, Figure S5A shows the experimental protocol of the rheological characterization and Figure S5B shows the related mechanical response of the collagen network. Here, it can be seen that the plastic behaviour is conserved with variation of the strain amplitude. Further, we removed the data for Matrigel – originally, we thought it would be instructive to see the rheological difference between these two model systems of ECM, but do agree that this is a bit beyond the scope of the current manuscript.

Changes were made: Fig. S5

10. The methods section of the manuscript does not contain enough information for an independent researcher to reproduce the experiments. For example, the immunofluorescence section should include antibody dilutions used. In addition, the methods section should include information about the concentration of Cytochalasin D (only listed in description of video S5), marimastat (listed at various points in the manuscript but not in the method section), and Y27632.

We thank the reviewer for the request for additional information and extended the Material and Methods section as suggested. We added the dilution of each antibody to Table S2-S4. Additionally, we added a new paragraph to the Material and Methods in which we describe the inhibitor treatments more precisely including concentrations and diluents.

Changes were made: Materials and Methods; Additional data: Table S2-S4

11. The article could be improved by discussing the relevance of the findings to *in vivo* mammary epithelial branch elongation. Is the concept of a collagen cage relevant for *in vivo* branching morphogenesis? Is the concept of ECM plasticity directing branch elongation consistent with the pattern of the mammary epithelium *in vivo*?

We agree with the reviewer and extended the discussion with the *in vivo* relevance of our findings, focusing on the collagen cage, fiber alignment and role of E-cadherin. Indeed, collagen structures observed *in vivo* are reminiscent to the structures found here *in vitro*. In line with our observations of collagen accumulation around growing branches, it has been shown that the murine mammary gland is covered with a dense layer of collagen, which constrains its expansion in the final stages. In our assay this coverage and the resulting constrain defines the branching behaviour by allowing branch initiation solely at the branch tip. Further, it has been shown that *in vivo* branch expansion is facilitated by aligned collagen fibers within the mammary fat pad, which can be generated by stromal cells and the expanding epithelium. Here, our model sheds light on the intrinsic and collective tension build-up of the extending epithelium, leading to plastic fiber alignment.

Changes were made: Manuscript pg. 1, pg. 6

Additional comments that will help improve clarity and overall quality of the article:

1. Please consider avoiding the use of red and green in schematics (e.g. Figure S1D). These figures are difficult to see for anyone who is red-green colorblind. Red can be replaced with magenta or gray to avoid this problem. Here is the link for a helpful article: <https://www.ascb.org/science-news/how-to-make-scientific-figures-accessible-to-readers-with-color-blindness/>

We thank the reviewer for this valuable recommendation. As suggested, we replaced the colours red and green in Fig. 2B, S1C and S1D by red and cyan.

2. Error bars should be added to Figure 1E. Otherwise, it is unclear how reproducible the observed decrease in displacement is.

According to the suggestion, we added error bars to Figure 1E.

3. The phrase “Analogous, also branch elongation” on pg 3 of the manuscript is unclear and should be reworded.

We reworded the sentence to read now: “In addition, branch elongation occurred discontinuously in time, showing a back-and-forth movement of the leading cells (Fig. 1H, grey line). Together, these observations hinted towards dynamic cellular rearrangements within elongating branches. “

4. The sentence “In order to identify, the mechanism...” on page 3 does not need a comma.

We thank the reviewer for this observation and corrected this mistake.

5. The labels for LifeAct and Nuclei in Figure 2A and D, respectively, are hard to read. Please consider reformatting.

For better readability we changed the line style and colour of the labels in Figure 2A, 2D and S4D.

6. In Figure 2D and E, it is hard to compare the data because the velocity is reported as arbitrary units in Figure 2D and as $\mu\text{m}/\text{min}$ in Figure 2E. Consider using consistent units for the entire figure to enable a comparison.

We restructured the figure according to the reviewer's comments. We plotted the vector field of the internal cell migration and further separated the velocity in a parallel and orthogonal component. We plotted the according velocities in the same units.

Changes were made: Fig. 2D, 2E

7. In Figure S2D the magenta channel seems to show nuclei and the position of the beads. Adjust the label accordingly.

We thank the reviewer for this comment and changed the legend accordingly.

8. In Figure 3H, the authors refer to a “thin stick-like structure” while on page 4 of the text they refer to “very thin and spindly branches structures”. Please be consistent with the description of the organoid morphology. Otherwise, it will confuse some readers.

For better comprehension we changed the description in the manuscript to “thin stick-like structure”.

9. The order of figure panels in Figure S3E-H does not match the text. Also consider showing expression of MMP9 first, before showing the results of blocking MMPs during the establishment phase and the elongation phase to avoid confusion.

Indeed, showing the expression of MMP9 first facilitates understandability. We changed the order accordingly and rearranged the panels in Figure S3.

**10. On page 12 of the manuscript the authors write “according to the planned experiment, time”. The comma can be removed from this sentence.
11. On page 13 of the manuscript, the authors write transduced and after 24hours”. Space missing between 24 and hours.**

The comma has been deleted and the space bar was added.

**12. The scale bar value for panel B is missing the figure legend of Figure 4.
13. The scale bar value for panel D is missing from the legend for Figure S1.**

We added scale bars for both figures.

14. Y-27632 is referred to as Y27632 or Y-27632 throughout the paper. Please be consistent with the formatting.

We changed the nomenclature to Y-27632 throughout the whole manuscript.

Reviewer #2 (Remarks to the Author):

In this clearly written manuscript, Buchmann et al. use a 3D model of mammary epithelial branching in collagen to describe the tensile forces at play in the process of branching morphogenesis. Using live confocal imaging, the authors analyze the temporal and directional characteristics of cellular movement and ECM deformations and identify periodic collective movement of cells, leading to stable ECM remodeling which ultimately directs branch elongation. The authors nicely demonstrate the existence of a tension equilibrium between the elongating branch and the extracellular matrix, and describe the formation of a stable collagen cage that encapsulates the epithelial branch.

Overall, while this work uses a compelling model to characterize tensile forces enacted on the ECM by epithelial cells and vice versa, most of the observations described are not novel. In addition, only a superficial description is made on observations that are further investigated, and no mechanism is provided.

We thank the referee for pointing out the importance and relevance of our work. Indeed, our organoids are derived from individual human primary cells, giving rise to self-organized structures which resemble very closely the tissue architecture of the human mammary gland. We are very proud of the quantification and in-depth analysis of the growth process, which has not been reported before and will pave the way for further progress in the field.

While it has been merely observed that the cells interact with the collagen matrix, the present work now is the first to unambiguously demonstrate that the interaction between cells and matrix leads to the formation of a stable collagen cage, which in turn is prerequisite for the observed collective cell migration resulting ultimately in the plastic deformations of the ECM. Our results summarize into a general mechanical tension model of the self-organizing process which provides a basis for addressing even more complex morphogenetic processes.

The referee comments helped us to improve the writing to clarify that our results go beyond normal branching morphogenesis but might also be relevant for invasive processes as observed during breast cancer progression. Therefore, this study provides an important basis to look at the role of plastic deformation and collagen cage formation not only in morphogenesis but also to determine how this process is coopted during tumorigenesis.

Data presented here that is not novel:

1. The need for Rho-kinase for branching morphogenesis (described in ref 21)

Of course, Rho-associated kinases participate ubiquitously in cellular processes requiring actin reorganization and thus, whenever a cell moves. This observation alone is not a major point of the manuscript, yet important to show that ROCK-dependent intrinsic contractility is a prerequisite for the observed tension buildup which in turn leads to the formation of the collagen cage. These analyses also serve to reveal that tension buildup is generated by the collective motion of cells, integrating individual traction forces for tension buildup. In this context, intrinsic contractility as a prerequisite for sufficient tension buildup to generate a stable collagen cage is to the best of our knowledge indeed a novel insight into branching morphogenesis of mammary glands.

2. The permanent nature of the ECM changes as a result of cell movements (described in ref 19 and by others)

Indeed, it has been described previously that contractile cells can plastically remodel collagen networks. In fact, we discussed this point in the introduction and discussion where we cite for example “Kim, J. et al. Stress-induced plasticity of dynamic collagen networks. Nat Commun 8, 842 (2017)” showing that individual cancer cells are able to mechanically reorganize the ECM. However, in our model system plastic and long-ranging deformations of the ECM are a collective effort by multiple cells only enabled by the integration of the individual traction forces of cells within branches. As stated above, to our best knowledge this combination of collective back-and-forth migration and plastic deformation is indeed novel and has not been described elsewhere. Furthermore, the plastic behaviour not only leads to the plastic collagen fibre alignment, but also the build-up of a stable collagen cage, which is needed to guide the collective motion. A description of such a collagen cage, resulting in a mechanical feed-forward loop has also not yet been described. In order to further reconcile our findings with previous studies and to highlight their novelty we extended the introduction and discussion.

Changes were made: Manuscript pg. 1, pg. 5

3. The role of collective cell migration in remodeling the ECM through tensile forces (described in ref 9 and others)

Reference 9 uses rectangular prepatterned collagen networks to observe the extension from branch-like structures from the edges of murine cell clusters. The patterning is based on a network of 6 mg/ml collagen, which limits the strain and the growth of full branching structures. While the paper nicely shows that biophysical forces are critical for 3D collective migration, the limitations of the experimental setup did not allow to resolve the migration track of individual cells during collective cell motion, nor the outgrowth of branches or the formation of a collagen cage. While clearly Ref. 9 and others offer important insights which are the basis for current progress, the present work integrates isolated observations into one coherent model of branch elongation and does so in a human organoid model based on primary cells. It is not merely the point that collective cell migration remodels the ECM, but the importance of the intricate balance between invasion, ECM remodeling and the collective movement of the cells, which we highlighted in the introduction and discussion.

Changes were made: Manuscript pg. 1, pg. 5

Findings presented here that were not further investigated:

1. The periodic nature of extension and retraction of cell movements that correlate with ECM deformation: this was previously observed in ref 9, but was not elaborated on there, either. Here, it was more thoroughly described, but there was no attempt to test the necessity of this periodic nature for ECM remodeling or branching morphogenesis, or to find the underlying mechanism behind the periodic collective cell movement.

The reviewer raises an interesting point that, however, we strongly feel goes beyond the current scope of the manuscript. To explore the mechanism behind the periodicity of the extension and retraction of cell movements requires a host of completely new experiments

including a much more in-depth characterization of these oscillations that together warrant inclusion in a future manuscript.

2. Tip cell overtaking: this is a recorded phenomenon in endothelial branching (Boas and Merks, BMC Syst Biol 2015), but its significance or underlying mechanism are not known. It is clearly demonstrated and comprehensively described here, but not investigated or discussed further. It would be interesting to know if it is entirely stochastic and whether it plays a role in the branching/elongation process.

We thank the reviewer for this useful suggestion. First of all, it is important to note, that albeit observed in endothelial branching, there is no report of such dynamics in mammary gland branching morphogenesis. However, according to the reviewer's useful suggestion, we analysed the process of tip exchanges in more detail. As a consequence, we observed that a tip exchange was slightly more probable in shorter and narrower branches. Further, we saw in more than 70% of exchanges that the tip cell stopped its initial migration prior to an exchange. By contrast, stalk cells were migrating outwards in more than 70% of exchanges. We added those observations to the manuscript and the related quantifications to the supplements. Taken together, we conclude that specification of tip cells is not entirely random, but appears in an ordered manner that is initiated by deceleration of the tip cells.

Changes were made: Manuscript pg. 3; Additional data: Fig. S2

3. The formation of a stable collagen cage: the authors describe the collagen cage, but only assume the nature of its formation through tensile forces. Experimental interventions are needed in order to understand:

i) the role of collagen secreted by the cells vs. pre-existing matrix collagen in the formation of the cage (laminin is shown to be secreted by the cells, was collagen secreted too?)

As this is a very important point, we agree with the reviewer to further discuss the components of the cage and their origin. To this end, high-resolution imaging of the cage and the microstructure of the collagen gel was realized by cultivating the cells inside a mix of fluorescently labelled and non-labelled collagen throughout the whole development of the organoid (Fig. 4B). As a consequence, the cage we observed was build-up from pre-existing collagen. Additional immunofluorescence imaging during alveologenesis phase revealed laminin to be an additional component of the cage (Fig. 4G). However, the additional new experiments did not detect any other components of the basement membrane such as Collagen IV or fibronectin.

Changes were made: Manuscript pg. 5

ii) the role of the cage in preventing or directing side branching as well as directional elongation through the branch tip (It would be interesting to manipulate areas of the cage to see how it affects branching and elongation)

We thank the referee for raising this point to strengthen our major claims. To further analyse the function of the collagen cage, we characterized the side branching behaviour of the organoids. In total we analysed 61 branching events of 26 organoids of three different donors. Interestingly, we observed side branching only via bifurcation of the tip

of the branches, while new branches never arose at the side of already existing branches. This further underlines that the collagen cage acts as barrier to prevent cells invading the ECM perpendicular to the branch direction. We added a time series of a branching event to the supplemental material (Fig. S6). Indeed, these results are in line with *in vivo* data on morphogenesis in the mouse mammary gland, where bifurcation of the branch tip drives epithelial expansion during puberty.

Changes were made: Manuscript pg. 5, pg. 6; Additional data: Fig. S6

iii) other roles the cage may play in mediating epithelial/ECM interactions. Importantly, as this finding is among the highlights of this work, it seems essential to demonstrate how a collagen cage formed *in vitro* relates to branching morphogenesis *in vivo*: is there evidence of an equivalent matrix cage in the breast? How does an *in vitro* collagen cage correspond with *in vivo* basement membrane?

Indeed, we do see a strong overlap of our *in vitro* observations and previously described *in vivo* data. On the one hand, it has been shown *in vivo* that aligned collagen fibers, present in the mammary fat pad, guide branch orientation and elongation, similar to the fibre alignment field in our assay, which originates from the epithelium itself. On the other hand, similar to the collagen cage in our assay, established murine mammary ducts are surrounded by a dense layer of collagen. It is proposed that this layer stabilizes the duct and constrains epithelial expansion. In our assay, the collagen cage consists of collagen I, the major structural component of the human mammary ECM and is lined with laminin, a major component of the basement membrane. However, we did not detect collagen IV or fibronectin. Finally, we extended the discussion and added additional references.

Changes were made: Manuscript pg. 1, pg. 6

Specific comments:

• **Most of the information presented in S1A-D was previously shown in Linnemann, Development 2015.**

Indeed, we use the organoid model first described by Linnemann et al., however, we greatly expand on it in this manuscript. Figures S1A-D, as mentioned by the author, do not merely repeat what has been shown in the first paper describing this experimental system, but go into much greater detail. Specifically, the temporal lineage marker expression, the cell-matrix adhesion via integrin and the overarching cellular actin organization provide a solid basis for our studies into cell-ECM interactions.

Additional data: Fig. S1D

• **Fig. 1G: it is difficult to tell the two brown hues apart in the graph.**

For clarity we additionally changed the line styles inside the figure and extended the caption.

Reviewer #3 (Remarks to the Author):

The manuscript titled “Mechanical plasticity of the ECM directs branch elongation in human mammary gland organoids” presents an interesting characterization of how a balance of cell and ECM tension, along with cyclic collective cell motility, drives large scale rearrangement of the ECM to form a collagen cage and promotes ductal branch elongation and invasion. The strengths of the study include a new and relevant experimental model (primary human mammary organoids derived from single basal cells in detached collagen gels), quantitative imaging of the organoids and surrounding ECM, including timelapse microscopy. The finding that cyclic changes in the direction of collective motility (periodic out/in behavior) can drive large scale changes in ECM alignment is particularly exciting and sheds new light into the dynamical mechanisms by which organoids remodel the ECM and ducts elongate. The study can benefit from additional characterization of the organoid system and how cellular tension and cell cortex coupling (adhesion-mediated) affects the cyclic nature of cell movement. Further discussion on how this body of work fits the existing literature on mammary gland morphogenesis (both *in vivo* and *in vitro*), as well as model system of collagen remodeling in 3D culture, will be helpful.

We thank the reviewer for pointing out the relevance and novelty of our work. Following the suggestions, we further characterized the organoid system, specifically the temporal and spatial expression of lineage markers. Additional experiments were conducted in order to more carefully examine the role of cell-cell adhesion in tension generation and cell movements. We edited the discussion to highlight the *in vivo* relevance of the here described model system and the resulting observations.

Main comments:

***Because this is a relatively new model system, it’s important to report on how efficient each stage in the organoid morphogenesis is, and provide more details on how specific organoids are selected for analysis. Important points to describe include: What is the efficiency with which single cells establish the stage 1 organoids? What is the frequency of transition between stage I/II; stage II/III? What is the frequency with which GATA3(+) organoids form from stage I organoids? Is the time point of GATA3(+) cell emergence stereotyped? If so, what is the structural event that correlates with GATA3 cell emergence? Without this information, it would be near impossible for another lab to evaluate whether they were successfully reproducing the results. Further, it would make building on the work challenging.**

In this assay 1 out of 400 basal cells can form a TDLU-like structure as described previously by Linnemann et al. Such a cell forms a small, but already branched cell cluster during the end of the establishment phase at day 6. Once such a cell cluster is formed, we observe those structures to fully undergo branching morphogenesis and alveologenesi. Yet, it has to be noted that alveologenesi is limited to freely floating collagen gels and does not take place when collagen gels are attached to the culture dish. The emergence of GATA3(+) cells can be observed around day 10 when an inner cell layer has formed. In order to show the temporal expression patterns of GATA3 we added additional immunostaining to Fig. S1D. In conclusion, we think that this additional information is indeed important for the readers.

Additional data: Fig. S1D

***The fact that basal/myoepithelial cells express both E-cad and P-cad -- the latter expressed at higher levels -- might suggest that the role of E-cad in these structures is more complex than simply coupling cell contractility or cortices. How does cell adhesion/coupling of cell cortices affect coordinated cell movement? What is the role of P-cadherin in these movements? For example, does the E-cad blocking antibody (or P-cad) reduce cell coupling which reduces the coordination between cells and impairs branch elongation.**

We thank the reviewer for this highly interesting and constructive idea. To follow up on it, we further probed the role of E-cadherin in collective cell migration. Specifically, we performed live-cell imaging of developing organoids in the presence of HECD1, a function-blocking anti-E-cadherin antibody. Thereby, we observed that upon addition of HECD1, collective cell migration disappeared and instead, cells only underwent individual short-ranging cellular rearrangements. In addition, we observed the formation of filopodia perpendicular to the branch, something never detected in control organoids. Together, these data suggest that E-cadherin is necessary for collective migration of the stalk cells.

To further characterise cell-cell adhesion in extending organoid branches we performed immunostaining against p-cadherin (see Figure below). Thereby, we detected partial p-cadherin expression in 25 of 32 examined branches of in total 2 donors. Yet, we detected a strong uniform E-cadherin expression throughout all branches, indicating E-cadherin to be the major protein for cell-cell adhesion in our assay.

Changes were made: Manuscript pg. 4; Additional data: Fig. S3H, Video 7

***Aside from lacking alveologensis, which does not seem to be the focus of the rest of the manuscript, how different are the anti-E-cadherin-treated organoids during the branching morphogenesis stage? This seems like an important experiment for demonstrating the importance of collective tension, so it would support the argument if you included quantification of branching, cell motility, long-range ECM deformation, or ECM alignment for this condition. For example, is collective tension dispensable or necessary for the formation of a collagen cage or aligned fibers at the branch tip?**

Aside from lacking alveologensis, already during the branch elongation phase the morphology of the organoids differs drastically. Organoids treated with HECD1 become thinner, but longer and thereby look more spindle-like compared to untreated organoids. We added the quantification to Figure S3F/G.

Following the suggestion, we performed live-cell imaging of HECD1-treated organoids. We observed that the anisotropic long-ranging deformation field vanished. Only close to the edges of the freely floating collagen gel small deformations were detectable, as here the collagen gel was more compliant due to the boundary condition. However, in combination with the loss of collective cell migration these observations further emphasise our model, that collective tension build-up and collective cell migration is required for the anisotropic, long-ranging deformation field.

Since structures treated with HECD1 further elongate and cells invade the ECM we still detect significant fibre alignment in front of arising branches. However, the degree of fibre alignment is significantly lower than in control organoids (Fig. S4C).

To characterise the development of the collagen cage in dependency of E-Cadherin, we visualized the collagen network around structures which were treated with HECD1. We observed that those structures showed significantly less collagen coverage than control organoids (see figure below), highlighting once again the necessity of E-cadherin mediated cell-cell adhesion in tension generation and matrix deformation.

Changes were made: Manuscript pg. 4; Additional data: Fig. S3,S4C, Video 7

p-Value: 2.3207e-18.

Minor comments:

***The discussion of “small range deformations ... due to the invasion dynamics of the leading cells ... asynchronous to the large deformation field” seems to be missing a transition to “Thus, the observed fiber alignment was a direct consequence of the observed contractile deformation field”. The purpose of the paragraph seems to be that, despite short-range deformations based on the invasion dynamics of the leader cells, the overall fiber alignment is caused by the collective motion of cells. If so, then such an argument would be well-supported by a description of the collagen fiber alignment associated with treatments that do not affect the invasion behavior but do affect collective motion, like in the anti-E-cadherin experiment described in a previous section.**

We thank the reviewer for this astute observation. As described in the previous section, we treated organoids with HECD1 to reduce cell-cell adhesion. As a consequence, we observed that collective cell migration and the long ranging deformation field vanished (Fig. S3H), highlighting once again the cohesion between collective cell migration and deformation. Further, by analysing the alignment field in front of growing branches, we found significantly less fibre alignment in front of invading branches than in control

organoids, underlining the importance of collective tension build-up for branch elongation (Fig. S4C).

Changes were made: Manuscript pg. 4; Additional data: Fig. S3, S4C, Video 7

***In discussion, “at the front the thinner collagen cage permits collagen degradation by the leading cells” is not quite congruent with the results, which instead point to “at the front the thinner collagen cage is associated with collagen degradation by the leading cells”.**

We thank the reviewer for this correction and changed the text on page 6 accordingly. It now reads “... at the front the thinner collagen cage is associated with collagen degradation by the leading cells ...”.

***The manuscript is a detailed characterization of invasive branching morphogenesis of mammary organoids in a collagen gel. The introduction references branching morphogenesis in development, and the discussion would be more impactful if it could compare the observations in this work with the existing literature on the process of mammary branching morphogenesis, both observed in vivo or in other mammary organoids.**

We agree with the reviewer and extended the discussion to put our results in the context of previously described findings *in vivo* and *in vitro*. We do see a distinct overlap between our observations and phenomena described *in vivo*. In particular, aligned collagen fibers exist in the murine mammary fat pad and guide the epithelial outgrowth. Similar alignment patterns can be observed in our assay, where they are generated by the epithelium itself and orient further branch elongation. Our model helps to gain further insight on how the mammary epithelium can generate sufficient traction force to create an alignment field, which guides its own outgrowth. Ultimately, the arising collagen cage mimics what has been observed around murine mammary ducts, where a thickened layer of collagen has been observed.

Changes were made: Manuscript pg. 1, pg. 6

Figure 2

2A: Caption could be more descriptive, perhaps highlighting the actin-based protrusions or filopodia/invadopodia visible at the top edge.

We extended the caption, highlighting the dynamic behaviour of the invadopodia.

2B: For clarity, it may be helpful to rotate the images so that they have the same orientation as the branch in 2A (tip pointing up).

For clarification we rotated the images as suggested.

2C: Title is somewhat confusing because “events” are not “branches” (y axis). Could be rephrased as “Branches with exchanges within 24 hrs”

We replaced “Events counted within 24 hrs” by “Branches with exchanges within 24 hrs”.

2D: The separation of velocity into two separate graphs for the x and y component is somewhat confusing and may detract from the purpose of the panels (showing clusters of collective motion in the same direction). Could this be done in a different

way that is more clear, for example, by color-mapping angular direction or by plotting a vector field?

We thank the reviewer for this useful suggestion and restructured Figure 2D. Now, we first show the vector field of the cellular migration within an extending branch. In the further plot we split the velocities in a parallel and orthogonal component and color-coded the components.

2E and 2G have the same time scale, but seem to suggest different takeaways about the dynamics of the system. In 2E, it looks like there is a 200-min period of inward migration, followed by 100+ minutes of outward migration. In 2G and H, it looks like periods of inward and outward migration switch every twenty minutes or so.

We observed during its growth phase, every organoid develops a somewhat unique morphology as well as timing of its development. Specifically, we consistently see that branch length, width and number of cells varies and is dependent on organoid age, distance from the gel boundary and possible, distance from the next organoid. As a result, the spatial confinement provided by the cage is subject to variation between branches, leading to a varying cellular migration behaviour particularly with respect to the time scale and speed. In the case of Figure 2, we depicted two branches of different length, width and age to highlight that our observed phenomena are observed during all developmental stages and, despite individual differences in time scale, are generalizable for all organoids. Explicitly, the organoid shown in Figure 2E is 12 days old and the organoid in Figure 2G 8 days old, which we now state in the Figure caption.

Figure 3.

3H: Could the legend labels (TLDU-like, Star-like, Thin stick-like) be vertically distributed to align with the matching inset images? This makes it more clear that these images are intended to complement the legend.

We thank the reviewer for this suggestion and changed the legend accordingly.

Some figures may be inaccessible to people with colorblindness, such as Figure 2 and Figure S1, which use red and green.

We thank the reviewer for pointing out this important issue. We replaced the colours red and green in Fig. 2B, S1C and S1D by red and cyan.

Minor comments (text):

“long ranging deformation” should be “long-range deformation”
“We observed a high expression” should be “We observed high expression”
“A phalloidin staining” should be “Phalloidin staining”
“Accordingly, the ECM was relaxing” should be “Accordingly, the ECM relaxed”
“Thus, the tension is built up by the whole branch and is equilibrated by the surrounding ECM” seems to belong after the following paragraph, after discussing the anti-E-cadherin antibody.
“collective tension built-up” should be “collective tension buildup” here and in other places where “built-up” is used as a noun rather than an adjective
“Here, the alignment of fibers showed to decrease with increasing angle” should be “Here, the alignment of fibers decreased with increasing angle”
“Each cycle, leads to plastic fiber elongation” should not have the comma there

**“Leading to only a localized but spatiotemporal inhomogeneous deformation field”
might be intended to be “leading to not only a localized but also a spatiotemporally
inhomogeneous deformation field”
“We observed that this tension induced collagen fiber alignment” should be “We
observed that this tension-induced collagen fiber alignment”
“While at the side the dense cage prevents a further outgrowth” should be “While
at the sides the dense cage prevents further outgrowth”**

We thank the reviewer very much for the detailed and very constructive corrections. We changed them in the manuscript accordingly to the suggestions.

REVIEWER COMMENTS

Reviewer #1 (Remarks to the Author):

The authors have thoroughly revised the manuscript and addressed most of the comments provided. New experiments and quantifications added strengthen the conclusions in the manuscript and improved data visualizations are helpful for clarity. The reviewer recommends minor revisions, which will help to more accurately describe how the results of the study relate to observations made in vivo and in previous studies of mammary organoids. Once these changes are complete, the manuscript will be suitable for publication.

1. The authors added comparisons of the collagen cage observed in culture to the collagen accumulation around mammary epithelial ducts in vivo, which help place the observations made in the manuscript into context. However, in the introduction of the revised manuscript, the authors state that “Such an encasing is similar to collagen densification observed around mammary ducts in vivo^{8,23}, which, based on our model, could result from epithelial self-organization.” Fibroblasts in the stroma are important for the production of this collagen layer, known as the periductal matrix, in vivo. The role of fibroblasts should be acknowledged. It can be misleading to suggest that epithelial self-organization is responsible for the densification of collagen in the periductal matrix given that the mammary gland is embedded in a stroma that largely consists of adipocytes.

2. The authors state that “Further, this mechanical constrain limits branch initiation to the invasive front via bifurcation, as it is observed during the expansion of the mammary epithelium during puberty^{4,45}.” The authors should acknowledge that lateral branches are also formed during puberty (see review article: <https://doi.org/10.1186/bcr1368>) and that these branches extend through the collagen-rich periductal matrix surrounding the duct in a process induced by MMP3 (doi: 10.1083/jcb.200302090). The fact that MMPs are needed to break through the periductal matrix to make lateral branches is consistent with the authors’ arguments.

3. The authors state that “The aligned fiber bundles, in turn, exerted a restoring force to the branches, thereby generating tension equilibrium that ultimately enabled elongation of the branch. Similarly, in vivo, aligned collagen fibers have been shown to orient murine mammary epithelium outgrowth and it was proposed that such alignment is generated by stromal cells and the expanding epithelium itself^{8,23}.” How is the concept of tension equilibrium relevant to mammary epithelial branching in vivo? Terminal end buds are covered in a continuous basement membrane, and it is unclear how the cap cells that line the basal surface of these multicellular structures would experience tension as described in the culture model used in this study.

4. Introduction: The motivation and key findings of the study should be more clearly articulated. Sentences such as “However, in a three-dimensional environment, it remains a challenge to identify the underlying self-organizational principles driving branching morphogenesis.” do not clearly motivate the analysis conducted in this study. Try to be as specific as possible when identifying why this study was conducted and what the key conclusions of the study are. Where possible, include alternative hypotheses that you can differentiate between with the conducted experiments. This can be useful both in the introduction and in the results section. In addition, broad statements with many references such as “An essential cue is given by the structure of the ECM, as aligned collagen fibers suffice to guide the direction of branching epithelium^{8,9,16,17,18,19}.” are not informative and additional information is required so that readers can understand what these previous studies

found, what their limitations were, what was still unclear, and how they are related to the current study. The authors explained some of their points regarding the motivation of the study and the novelty well in the rebuttal letter (for example in response to reviewer 1 comment 5). Please include some of those statements into the introduction of the manuscript to help the reader understand the motivation and novelty assuming that many readers will ask themselves the same questions as all three reviewers.

5. Methods section: Thank you for clarifying that organoids were imaged 4 h after treatment with Cytochalasin D. This information and the duration of all other drug treatments should be added to the methods section of the manuscript so that other researchers can reproduce the data in the manuscript.

6. Figure S4C: Thank you for including the quantification of the degree of alignment of collagen fibers at the front and sides of the branches. The quantification nicely supports the author's conclusions. For the drug treatments, please clarify where alignment was measured (I assume at the front). Please also include significance as stars into the plot (in addition to listing it in table S5) and reference figure S4C in the main text in line 181 along with S4A and in line 184 along with S4B.

7. Figure S5: Thank you for adding the reference to figure S5 to the main text and for removing the Matrigel figure panel. Please update the methods section accordingly where it says "plastic behavior of the different hydrogels (Fig. S6)". In addition, Figure S5 should be referenced here instead of Figure S6.

Reviewer #2 (Remarks to the Author):

The authors did a nice job integrating the known understanding of tensile forces and how they operate in the mammary gland to direct epithelial growth in the ECM/stroma. It is a good model, useful, and puts things in a clear frame that does support further investigations.

The main concern remains about novelty. There is no major advance here, but several marginal ones. The main advance is the integration of mostly-known information into a new model.

Another remaining issue is still the claim that the collagen cage is directing ductal branching. Intuitively, it makes sense that the branches elongate via bifurcation and not side-branching because the sides are restricted by the dense collagen cage. But they do not experimentally show that this is indeed the reason for lack of side-branching (by showing, for example, that side branching occurs in the absence of a collagen cage). So it remains a compelling theory.

They did do further analysis, quantifying the bifurcation events and showing that no side-branching occurs, but no interventional experiments.

Reviewer #3 (Remarks to the Author):

Thank you for addressing our questions and comments

Reviewer #1 (Remarks to the Author):

The authors have thoroughly revised the manuscript and addressed most of the comments provided. New experiments and quantifications added strengthen the conclusions in the manuscript and improved data visualizations are helpful for clarity. The reviewer recommends minor revisions, which will help to more accurately describe how the results of the study relate to observations made in vivo and in previous studies of mammary organoids. Once these changes are complete, the manuscript will be suitable for publication.

We thank the reviewer for acknowledging the improvement of the manuscript by the extensive addition of experiments, data, analysis and clarifications in the first round of revisions. We are happy to see that the acceptance of our manuscript is recommended after including some minor changes. We included the suggested changes in the manuscript accordingly.

1. The authors added comparisons of the collagen cage observed in culture to the collagen accumulation around mammary epithelial ducts in vivo, which help place the observations made in the manuscript into context. However, in the introduction of the revised manuscript, the authors state that "Such an encasing is similar to collagen densification observed around mammary ducts in vivo^{8,23}, which, based on our model, could result from epithelial self-organization." Fibroblasts in the stroma are important for the production of this collagen layer, known as the periductal matrix, in vivo. The role of fibroblasts should be acknowledged. It can be misleading to suggest that epithelial self-organization is responsible for the densification of collagen in the periductal matrix given that the mammary gland is embedded in a stroma that largely consists of adipocytes.

We thank the reviewer for this hint and extended the introduction with additional information regarding the impact of fibroblasts during branch expansion.

2. The authors state that "Further, this mechanical constrain limits branch initiation to the invasive front via bifurcation, as it is observed during the expansion of the mammary epithelium during puberty^{4,45}." The authors should acknowledge that lateral branches are also formed during puberty (see review article: <https://doi.org/10.1186/bcr1368>) and that these branches extend through the collagen-rich periductal matrix surrounding the duct in a process induced by MMP3 (doi: 10.1083/jcb.200302090). The fact that MMPs are needed to break through the periductal matrix to make lateral branches is consistent with the authors' arguments.

We specified the discussion and added the suggested literature to the discussion.

3. The authors state that "The aligned fiber bundles, in turn, exerted a restoring force to the branches, thereby generating tension equilibrium that ultimately enabled elongation of the branch. Similarly, in vivo, aligned collagen fibers have been shown to orient murine mammary epithelium outgrowth and it was proposed that such alignment is generated by stromal cells and the expanding epithelium itself^{8,23}." How is the concept of tension equilibrium relevant to mammary epithelial branching in vivo? Terminal end buds are covered in a continuous basement membrane, and it is unclear how the cap cells that line the basal surface of these multicellular structures would experience tension as described in the culture model used in this study.

It is important to distinguish the dynamics and time course of the development towards the final structure and the final stage of end buds. While the latter can be observed *in vivo*, the former is impossible to observe *in vivo* in human mammary glands – due to obvious limitations. Our manuscript describes the dynamics of organoid formation during which the epithelium remodels its environment. Here, the described tension equilibrium is prerequisite for the necessary remodeling processes driving branch elongation during the organogenesis. During this phase, the collagen cage is first arising and mainly covering the organoid body but not the tip of the branch, which enables force transmission between the epithelium and the ECM overall.

The reviewer points rightly out, that after puberty round terminal end buds invade the fat pad and thereby are covered by a basal membrane. This is indeed what we observe in our organoid system during alveologenesis phase in which the branches round up and are fully surrounded by a thin layer of laminin. At this stage, we also observe the collagen cage to be fully closed around the whole organoid as best seen in Video 9. This final static structure is reminiscent to the basement membrane *in vivo*.

We now improved the text to clarify this important analogy of the final stage.

4. Introduction: The motivation and key findings of the study should be more clearly articulated. Sentences such as “However, in a three-dimensional environment, it remains a challenge to identify the underlying self-organizational principles driving branching morphogenesis.” do not clearly motivate the analysis conducted in this study. Try to be as specific as possible when identifying why this study was conducted and what the key conclusions of the study are. Where possible, include alternative hypotheses that you can differentiate between with the conducted experiments. This can be useful both in the introduction and in the results section. In addition, broad statements with many references such as “An essential cue is given by the structure of the ECM, as aligned collagen fibers suffice to guide the direction of branching epithelium^{8,9,16,17,18,19}.” are not informative and additional information is required so that readers can understand what these previous studies found, what their limitations were, what was still unclear, and how they are related to the current study. The authors explained some of their points regarding the motivation of the study and the novelty well in the rebuttal letter (for example in response to reviewer 1 comment 5). Please include some of those statements into the introduction of the manuscript to help the reader understand the motivation and novelty assuming that many readers will ask themselves the same questions as all three reviewers.

We thank the reviewer for the specific suggestions. We clarified the introduction, motivation and discussion accordingly.

5. Methods section: Thank you for clarifying that organoids were imaged 4 h after treatment with Cytochalasin D. This information and the duration of all other drug treatments should be added to the methods section of the manuscript so that other researchers can reproduce the data in the manuscript.

According to the reviewer’s suggestion, we extended the experimental description in the Materials and Methods section.

6. Figure S4C: Thank you for including the quantification of the degree of alignment of collagen fibers at the front and sides of the branches. The quantification nicely supports the author's conclusions. For the drug treatments, please clarify where alignment was measured (I assume at the front). Please also include significance as stars into the plot (in addition to listing it in table S5) and reference figure S4C in the main text in line 181 along with S4A and in line 184 along with S4B.

We thank the reviewer for this observation and specified the according caption and reference. Further, we specified the significance in Figure S4C.

7. Figure S5: Thank you for adding the reference to figure S5 to the main text and for removing the Matrigel figure panel. Please update the methods section accordingly where it says "plastic behavior of the different hydrogels (Fig. S6)". In addition, Figure S5 should be referenced here instead of Figure S6.

We changed the description in the Materials and Methods and rewrote the reference accordingly.

Reviewer #2 (Remarks to the Author):

The authors did a nice job integrating the known understanding of tensile forces and how they operate in the mammary gland to direct epithelial growth in the ECM/stroma. It is a good model, useful, and puts things in a clear frame that does support further investigations.

We thank the reviewer for acknowledging our efforts and our contribution to understand the interplay of tension and mammary gland organoid growth. Further, we thank the reviewer for highlighting the relevance of our model for further studies in the field.

The main concern remains about novelty. There is no major advance here, but several marginal ones. The main advance is the integration of mostly-known information into a new model.

Unfortunately, the reviewer does not specify the concerns of novelty, which we find quite surprising as we extensively explained in the first round the difference between single cell assays and other individual observations found in various organoid assays. Further, we highlighted the uniqueness of our model systems which relies on the culture of primary human cells and enables the generation of branched organoids that mimic the human mammary gland in architecture and function.

We are happy to see that the referee acknowledges the importance of our work and novelty of the comprehensive model. We allow us to refer to our extensive list and clarifications in the first respond round, which may have escaped the reviewer's attention. As pointed out, for some of our observations we transfer already described phenomena from 2D model systems, single cell assays and artificial systems to our model. However, it has to be raised that in most cases those processes have not been demonstrated in a self-organizing 3D epithelium before. Further, due to the *in vivo* relevance of our assay our observations highlight the importance of collective and dynamical tension generation in 3D systems, which are highly related to processes observed *in vivo*.

Another remaining issue is still the claim that the collagen cage is directing ductal branching. Intuitively, it makes sense that the branches elongate via bifurcation and not side-branching because the sides are restricted by the dense collagen cage. But they do not experimentally show that this is indeed the reason for lack of side-branching (by showing, for example, that side branching occurs in the absence of a collagen cage). So it remains a compelling theory. They did do further analysis, quantifying the bifurcation events and showing that no side-branching occurs, but no interventional experiments.

Indeed, we observe branching to happen predominantly *via* bifurcation and not *via* side branching. We thank the reviewer for suggesting analyzing this process further during the first round of revision. The dominant process of branching *via* bifurcation is already an important finding as it is also observed *in vivo* as reviewer 1 pointed out. As a result, our assay can be used to address remaining questions concerning the specific branching process, but which are beyond the scope of this manuscript. At no point of the present manuscript, we put forward an argument addressing the 'lack of side branching'. In addition, side branching is no major

point in proofing our tension-based elongation model, but an interesting observation fitting in our theory.

One major point we are showing in the manuscript is that the collagen cage is prerequisite for organoid formation. Consequently, without the self-organized collagen cage no side branching and no elongation would occur at all, making the description of side branching a chicken-egg dilemma. Nevertheless, we performed precise UV-laser ablation experiments cutting holes into the collagen cage and subsequently performed live-cell imaging. However, such experiments cause too substantial artefacts, which in turn induce cell death and make a reasonable interpretation questionable.

Unfortunately, the reviewer is not precise enough, that we could understand what he/she suggests in terms of experiments. Therefore, we refer to the comment of reviewer #1, who points out the important findings that indeed MMPs are needed to break through the periductal matrix to make lateral branches *in vivo*, which is very consistent with our arguments brought forward here. In order to clarify the aspect of side branching we adapted the text and added the references suggested by reviewer #1.

Reviewer #3 (Remarks to the Author):

Thank you for addressing our questions and comments.

We thank the reviewer for supporting the acceptance of the manuscript after our extensive and comprehensive response in the first round of reviews. We are very pleased to find that the referee agrees that our additional experiments, analysis and data have improved our manuscript significantly to make it acceptable.